# SPEED LIMITS FOR DEEP LEARNING

## ABSTRACT

State-of-the-art neural networks require extreme computational power to train. It is therefore natural to wonder whether they are optimally trained. Here we apply a recent advancement in stochastic thermodynamics which allows bounding the speed at which one can go from the initial weight distribution to the final distribution of the fully trained network, based on the ratio of their Wasserstein-2 distance and the entropy production rate of the dynamical process connecting them. Considering both gradient-flow and Langevin training dynamics, we provide analytical expressions for these speed limits for linear and linearizable (e.g. NTK) neural networks. Remarkably, given some plausible scaling assumptions on the NTK spectra and spectral decomposition of the labels– learning is optimal in a scaling sense. Our results are consistent with small-scale experiments with CNNs and FCNs on CIFAR-10, showing a short highly non-optimal regime followed by a longer optimal regime.

## 1 INTRODUCTION

While for most of its history, thermodynamics was concerned with describing systems near equilibrium, in recent years there have been breakthroughs in stochastic thermodynamics and our ability to describe far-from-equilibrium systems. Thermodynamic fluctuation relations, uncertainty relations, and speed limits (Crooks, 1999; Seifert, 2012; Vu & Saito, 2022; Benamou & Brenier, 2000) allow us to relate the equilibrium properties of systems to their non-equilibrium behavior. The thermodynamic speed limits in particular lower bound the time it takes a physical system's configuration to evolve from an initial to a final distribution; the bound is given by the *Wasserstein-2* distance in weight-space divided by the *entropy production* of the process. Applied to computation, such speed limits were recently used to show that modern CPUs can write bits within a $O(1)$ factor from the optimal writing rate (see more examples in Vu & Saito (2022)).

Far-from-equilibrium dynamical systems of great interest are trained neural networks. As their training can be thought of as a virtual physical process involving many degrees of freedom, it must also conform to the rules of thermodynamics. In particular, the training time obtained using Neural Tangent Kernel (NTK)-type dynamics (Jacot et al., 2018) or Langevin-type dynamics should be bounded by the thermodynamic speed limit. Given the costs of training large models, it is desirable to characterize the efficiency of neural networks from this perspective. In particular, understand the impact of various design choices and data-set properties on the speed at which neural networks can learn.

Here we address this gap and derive such time bound. Our main results are the following:

- We recast thermodynamic speed limits in deep learning terms showing, in particular, how entropy production relates to features of the loss landscape, the learning rate, and, for Langevin dynamics, the free energy.

- We derive analytical expressions for the Wasserstein-2 distance, entropy production, and the speed limit for linear regression and for DNNs trained in the NTK regime.

- Remarkably, we find that NTKs with a power law spectrum and an initial residue (the target minus initial prediction) having relatively uniform spectral decomposition exhibit optimal dynamics in the scaling sense. Namely, the actual speed is a $O(1)$ factor times the theoretically optimal speed limit. In contrast, for residues with a stronger power-law decaying spectral decomposition, this factor grows with the data-set size.

- We report a numerical study on CIFAR-10, showing both of the above behaviors. Interestingly, warm-starting makes the residues more uniform and puts us in the regime of optimal (up to $O(1)$ factors) learning.

**Related works** Several works draw connections between learning algorithms, thermodynamics, and optimal transport. Yaida (2018) studied stochastic gradient descent and derived stationary fluctuation-dissipation relations and used these as a measure of equilibration. Goldt & Seifert (2017b;a) analyze neural network learning of a binary classification rule and introduce a thermodynamic efficiency of learning. They then use this measure of efficiency to study the generalization capability of each neuron in the network. Other works show that SGD performs variational inference, but for the population loss with an entropy term (Mandt et al. (2017); Chaudhari & Soatto (2018); Mei et al. (2018)) drawing the connection to Wasserstein gradient flow Jordan et al. (1998). These works however do not address the relation to the initial distribution of the parameters and do not derive a bound on the training time.

## 2 Speed limits of learning

Consider a single neural network or an ensemble of such networks with weights $\boldsymbol{\theta} \in \mathbb{R}^P$ at initialization. Training the neural network for a duration $T$ could be viewed as a dynamical process, moving the initial distribution of network weights from $p(\boldsymbol{\theta}(0))$ to $p(\boldsymbol{\theta}(T))$. Generally speaking, thermodynamic speed limits provide lower bounds $T_{\mathrm{SL}} \leq T$ on the time it takes to perform such a process based on its irreversibility and the distance between the initial and final probability distributions of the learnable weights.

Speed limits have been derived for both discrete and continuous dynamical processes. Here we focus on two relevant continuous time processes, NTK-type dynamics and Langevin-type dynamics. Specifically, given training data, $\mathcal{D}$, and general loss function $\mathcal{L}(\boldsymbol{\theta}; \mathcal{D})$, we consider the Langevin algorithm described by the stochastic differential equation, with $\eta \geq 0$ being the learning rate

$$d\boldsymbol{\theta}(t) = -\eta \nabla_{\boldsymbol{\theta}} V(\boldsymbol{\theta}(t); \mathcal{D}) dt + \sqrt{2\eta\beta^{-1}} \, d\boldsymbol{\mathcal{B}}(t), \tag{1}$$

where $\boldsymbol{\mathcal{B}}(t)$ is a Brownian motion (unit variance random noise), with temperature (noise) $\beta \in (0, \infty]$, and for NTK-type dynamics we take $\beta^{-1} = 0$. We consider the initial condition for $\boldsymbol{\theta}(0) = \boldsymbol{\theta}_0$ distributed randomly as an independent Gaussian on all $\boldsymbol{\theta}_0$'s namely $p(\boldsymbol{\theta}_0) \propto e^{-\beta \|\boldsymbol{\theta}_0\|^2}$ [1]. The potential $V$ is given by

$$V(\boldsymbol{\theta}; \mathcal{D}) = \begin{cases} \mathcal{L}(\boldsymbol{\theta}; \mathcal{D}) & \text{NTK} \\ \|\boldsymbol{\theta}\|^2 + \mathcal{L}(\boldsymbol{\theta}; \mathcal{D}) & \text{Langevin} \end{cases} \tag{2}$$

For simplicity, considering Langevin dynamics, we focus on the case where training infinitely-long without the $\mathcal{L}(\boldsymbol{\theta}; \mathcal{D})$ term yields $p(\boldsymbol{\theta}(0))$. Furthermore, we keep the learning rate ($\eta$) implicit here, setting $\eta = 1$ in the following. Instead of doubling the learning rate, one can think of doubling $V$ and $\beta^{-1}$. The above equation is a continuum approximation of the dynamics of discrete gradient descent with white noise at a low learning rate.

### 2.1 Entropy production and irreversibility

The tendency of a process to evolve in a preferred direction in time is related to entropy. The second law of thermodynamics states that entropy cannot decrease over time. Conversely, entropy production relates to the probability of a process running forward in time compared to the process running backward in time.

To make this point operational, let $p(\boldsymbol{\theta}(0))$ denote the distribution of initial states and $p(\boldsymbol{\theta}(T)|\boldsymbol{\theta}(0))$ the conditional distribution that $\boldsymbol{\theta}(0)$ evolves into $\boldsymbol{\theta}(T)$ within time $T$. Likewise, $p(\boldsymbol{\theta}(T))$ is the distribution of the final state and the conditional distribution $q(\boldsymbol{\theta}(0)|\boldsymbol{\theta}(T))$ denotes the probability that the processes evolve from state $\boldsymbol{\theta}(T)$ back into the state $\boldsymbol{\theta}(0)$ within time $T$ along the path $\tilde{\boldsymbol{\theta}}(t) = \boldsymbol{\theta}(T - t)$. Entropy production (or irreversibility) is then defined by Seifert (2012)

$$R = \left\langle \ln \frac{p(\boldsymbol{\theta}(0))}{p(\boldsymbol{\theta}(T))} \right\rangle + \left\langle \ln \frac{p(\boldsymbol{\theta}(T)|\boldsymbol{\theta}(0))}{q(\boldsymbol{\theta}(0)|\boldsymbol{\theta}(T))} \right\rangle, \tag{3}$$

---

[1]Any variance changes across layers are implicit in the norm here

where the expectation is taken over the distribution $p(\boldsymbol{\theta}(0))$ of initial states, and their forward evolutions. The first term depends only on the initial and final distributions, the second term also encapsulates the dynamical process and its reversed process.

We next collect and combine various results for $R$ scattered in the literature and adapt them to three relevant machine learning settings. Without loss of generality, we take here the learning rate, $\eta = 1$.

Consider first the case of Langevin dynamics, one finds the simple expression

$$\beta^{-1} R = \beta^{-1} \ln \mathcal{Z}_T - \beta^{-1} \ln \mathcal{Z}_0 + \langle \mathcal{L}(\boldsymbol{\theta}(0)) \rangle, \tag{4}$$

where the so-called "free energies" $\beta^{-1} \ln \mathcal{Z}_t$ are related to $p(\boldsymbol{\theta}(t))$ via $p(\boldsymbol{\theta}(t)) = e^{-\beta V(\boldsymbol{\theta}(t))} / \mathcal{Z}_t$ at time $t \in \{0, T\}$. Notably, the irreversibility of the dynamical process depends only on the initial and final or current state.

Next, entropy production (Eq. (19)) can also be expressed as a dynamical quantity (Vu & Saito, 2022) from which we obtain (see Appendix A.2)

$$\beta^{-1} R = \int_0^T \langle \|\nabla_{\boldsymbol{\theta}} V\|^2 \rangle - 2 \beta^{-1} \langle \Delta_{\boldsymbol{\theta}} V \rangle + \beta^{-2} \langle \|\nabla_{\boldsymbol{\theta}} \ln p\|^2 \rangle \, dt. \tag{5}$$

Note that, $R$ depends on $T$ for finite time $T$. In the low noise limit, $\beta \gg 1$ the first term dominates, which has the simple interpretation of the average squared length of the gradient. The next leading term contains the average Hessian of the loss function.

Finally, turning to the NTK case ($\beta^{-1} \to 0$) we find that at any time $T$ during the dynamics

$$\beta^{-1} R = \left\langle \mathcal{L}(\boldsymbol{\theta}(0)) - \mathcal{L}(\boldsymbol{\theta}(T)) \right\rangle. \tag{6}$$

## 2.2 Speed limits from optimal transport

The evolution of weights can also be phrased as an optimal transport problem. In particular, the operation of transporting initial weights to final weights could be described by the joint probability distribution $p(\boldsymbol{\theta}(0), \boldsymbol{\theta}(T))$ whose two marginals are the initial and final/current distributions. This joint probability also called a *plan*, can be thought of as the chance of $\boldsymbol{\theta}(0)$ to end up in $\boldsymbol{\theta}(T)$ by some process. One can then define the *cost* of a plan and ask what is the optimal plan. One relevant cost function to consider is the Euclidean distance-squared $\langle \|\boldsymbol{\theta}(0) - \boldsymbol{\theta}(T)\|^2 \rangle$. The Wasserstein-2 distance between the initial and final distribution, $\mathcal{W}_2(p_0, p_T)$ is defined as the minimal value of this cost when optimized over all possible plans (Eq. (27) in Appendix A.3).

The dynamical process itself yields a specific plan ($p(\boldsymbol{\theta}(0), \boldsymbol{\theta}(T))$). Remarkably, it turns out that $T\beta^{-1} R$ is equal to the cost of the plan $p(\boldsymbol{\theta}(0), \boldsymbol{\theta}(T))$ (see Appendix A.3 for details). Noting next that this plan cannot be more optimal than the plan underlying $\mathcal{W}_2(p_0, p_T)$ (i.e. $T\beta^{-1} R \geq \mathcal{W}_2(p_0, p_T)$) yields the thermodynamic speed limit known as the Benamou–Brenier formula (Benamou & Brenier, 2000; Vu & Saito, 2022)

$$T \geq T_{\text{SL}} \equiv \frac{\mathcal{W}_2(p_0, p_T)}{\beta^{-1} R}. \tag{7}$$

Besides obtaining $R$, as discussed in the previous section, the above formula requires solving the optimization problem underlying $\mathcal{W}_2(p_0, p_T)$. While this can be difficult in general, exact formulas are known for the Gaussian distribution, Dirac delta distributions, and one-dimensional distributions. In particular, considering a well-defined initial and final state $p_x(\boldsymbol{\theta}) = \delta(\boldsymbol{\theta} - \boldsymbol{\theta}_x)$ for $x \in \{0, T\}$, the Wasserstein distance $\mathcal{W}_2(p_0, p_T)$ simplifies to the $L_2$ distance $\|\boldsymbol{\theta}_T - \boldsymbol{\theta}_0\|^2$ in weight space. Otherwise, various useful bounds exist, (Cuesta-Albertos et al., 1996) as well as promising deep-learning-based numerical techniques (e.g. Courty et al. (2017); Kolouri et al. (2020); Genevay et al. (2016); Staib et al. (2017); Taghvaei & Jalali (2019); Korotin et al. (2020); Peyré et al. (2017)).

### 2.2.1 Meaning of the Speed limit and entropy production in deep learning

The speed limit involves a lower bound on training time, the distance between initial and final probabilities, and entropy production. Given a fixed entropy budget, it then bounds the time it takes

to perform this process for any Langevin dynamics, including dynamics with different and time-dependent potentials. Entropy, despite being a pillar of thermodynamics and many-body physical phenomena, is not a frequently measured quantity in deep learning. Consequently, it is desirable to explain some consequences of entropy production and hence the speed limit.

The simplest setting is gradient flow with a time-independent potential, where the free energy $(\beta^{-1}R)$ coincides with the decrease in train loss $(\Delta\mathcal{L}(T) = \mathcal{L}(\boldsymbol{\theta}_0) - \mathcal{L}(\boldsymbol{\theta}_T))$. Furthermore, the learning rate can be absorbed into the scale of the training loss, hence higher learning rates would imply higher entropy production. An optimal training of the DNN then has several related merits: **(i)** Given a fixed $\Delta\mathcal{L}(T)$ budget, no better loss function that takes us between the initial and final state can make the network travel this distance in weight space quicker. So, for instance, if we saturate the speed limit, no benefit can be gained by taking the Mean square error (MSE) loss to be $L_1$ loss or taking any other surrogate loss function (Nguyen et al., 2009; Yuan et al., 2021). **(ii)** For a fixed initial condition, where $\mathcal{W}_2(p_0, p_T)$ distance becomes $L_2$ distance, the network weights travel along straight lines in weight space, i.e. $\|\boldsymbol{\theta}_0 - \boldsymbol{\theta}_T\|^2$. Furthermore, the drop in train loss along the path is $\Delta\mathcal{L}(T)/\mathcal{W}_2(p_0, p_T)$ where $l$ is the distance along the path. Hence, entropy production is uniform in the distance along the path. We note that from a practical point in this noiseless case, one calculates the speed limit efficiently since it amounts to

$$T_{\text{SL}}(\beta \to \infty) = \frac{\|\boldsymbol{\theta}_0 - \boldsymbol{\theta}_T\|^2}{\Delta\mathcal{L}(T)}. \tag{8}$$

We next address the notion of a loss budget, relevant to point (i) above. Indeed, the scale of the loss may appear arbitrary and, if so, one can scale up the loss or the learning rate, such that the implied time-bound goes to zero. Within our continuum description, this is indeed the case, and scaling up the loss would simply speed up the dynamics and scale down the time-bound in a proportional manner. However, as far as our description mimics gradient descent (GD), one can only consider small gradients and hence a small loss/learning rate. At higher gradients, discrete GD would start deviating from its continuum approximation, and at even higher learning rates it often leads to NaNs. Analogously to how the binding energy of an atom sets a meaningful energy scale in physics (electron volt), these discrete effects, which depend on model and training choices, set a scale to the loss. The speed limit, as derived from the continuum, implies nothing about this scale. Still, given that we are well below this scale, it bounds the speed of gradient-flow dynamics. In principle, other speed limits relevant to discrete dynamics could be derived based on similar models (Vu & Saito, 2022).

## 3 CASE STUDIES

Here, we present two examples where the bound can be evaluated analytically. The first example illustrates the interplay between the speed limit, entropy production, and noise in the algorithm for the linear regression model (or linear perceptron). The second example illustrates how the optimality in training is related to the structure of the spectrum of the NTK, as well as the discrepancy from the target.

### 3.1 LINEAR REGRESSION - IN HIGH DIMENSION

Consider the problem of linear regression with scalar output, given a dataset $\mathcal{D}_n = \{X, \boldsymbol{y}\}$ where $X \in \mathbb{R}^{d \times n}$, and $\boldsymbol{y} \in \mathbb{R}^n$. The output of the algorithm is $\hat{y}(\boldsymbol{x}) = \boldsymbol{\theta}^{\text{T}}\boldsymbol{x}$, where the weights $\boldsymbol{\theta} \in \mathbb{R}^d$ are learned via Langevin dynamics (Eq. (1)). We consider the squared error loss $\mathcal{L}(\boldsymbol{\theta}; \mathcal{D}) = \frac{1}{2}\|\boldsymbol{y} - X^T\boldsymbol{\theta}\|^2$ with weight decay, and initial distribution $\boldsymbol{\theta}_0 \sim \mathcal{N}(0, (\lambda d)^{-1}I_d)$. In this case, since the initial and final distribution are Gaussian distributions, one can provide exact equations for the dynamics of $\mathcal{W}_2(t)$, and $R(t)$ for any noise level $\beta$, see Appendix C for details of the derivation.

To gain intuition, below we explore the speed limit bound in the asymptotic regime where the number of samples $n$, commensurate with the input dimension size, $d$, such that, $d/n \to \gamma \in (0, \infty)$, while $d, n \to \infty$. To facilitate the analysis, we assume a teacher-student setting with the target model $y = \boldsymbol{\theta}_\star^{\text{T}}\boldsymbol{x}$, with the true weights distributed, $\boldsymbol{\theta}_\star \sim \mathcal{N}(0, \alpha/d I_d)$. In addition, we assume that $X$ has i.i.d. entries. In Appendix C we provide an exact formula for the speed limit bound, $T_{\text{SL}}(\lambda, \beta, \gamma, \alpha)$, which depends only on these four parameters, noise level, $\beta^{-1}$, the variance of the true weights, $\alpha$,

initial variance intensity, $\lambda$, and the limiting dimension ratio, $\gamma$. There are a few interesting limits one can explore. First, taking the limit of $\beta \to \infty$ i.e. zero noise (gradient descent), the speed limit amount to a specific number,

$$T_{\text{SL}}(\beta \to \infty) \to 2 \frac{1 + \alpha\lambda}{\int s \, d\rho(s)}, \tag{9}$$

with $\rho$ being the Marchenko-Pastor distribution i.e. the limiting eigenvalues' distribution of the co-variance matrix $XX^{\text{T}}/n$. See Appendix C for more details. The main source of entropy production in this limit is the loss at initialization.

Second, taking the opposite limit of high noise we have,

$$T_{\text{SL}}(\beta \to 0) \to 0. \tag{10}$$

In this regime, the system is driven by the noise, and essentially the distribution at the end of training is equal to the distribution at initialization, therefore one can learn at zero time. We note that the speed limit is a lower bound. Moreover, it has no direct implication on the ability of the estimator to generalize.

Third, in the large samples' regime, $n \to \infty$, $(\gamma \to 0)$ corresponds to $n \gg d$, the bound reaches the following finite value:

$$T_{\text{SL}}(n \to \infty) \to 2\lambda\alpha. \tag{11}$$

This limit is in essence where the training error reaches the population error, hence the dynamics is in essence with respect to the population error. Remarkably, it is independent of the noise level $\beta$.

Last, in the over-parametrized regime, $d \to \infty$, and $d \gg n$ $(\gamma \to \infty)$ we have that

$$T_{\text{SL}}(d \to \infty) \to 0. \tag{12}$$

Interestingly, in this regime, the parameters are not moving a lot and therefore the final distribution is very close to its initial one. This is not the case when the noise is zero, as follows from Eq. (9). Therefore, the limit of $d \to \infty$ does not commute with the limit of $\beta \to \infty$.

We note that Eq. (9) and Eq. (11) show that even in the limit of zero noise and infinite samples, what makes the learning slower is high regularization or low variance of initial weights and high variance of the true target weights.

## 3.2 NEURAL TANGENT KERNEL (NTK) DYNAMICS

As a second analytically tractable example, consider a neural network trained in an NTK setting from a given fixed initial state ($\boldsymbol{\theta}_0$) for some time $T$. As no noise is introduced, $T$ determines the final state ($\boldsymbol{\theta}_T$) and decrease of the training loss ($\Delta\mathcal{L}(T)$). Consequently, one can think of the time-bound here as a function of $T$. We define inefficiency via the ratio $T/T_{\text{SL}}(T) \geq 1$. Specifically, it is given by

$$\frac{T}{T_{\text{SL}}(T)} = \frac{T\beta^{-1}R}{\mathcal{W}_2} = \frac{T\Delta\mathcal{L}(T)}{\|\boldsymbol{\theta}_0 - \boldsymbol{\theta}_T\|^2}. \tag{13}$$

The NTK dynamics, being linear, lends itself to exact analytical expressions for all quantities involved. Specifically,

$$\|\boldsymbol{\theta}_0 - \boldsymbol{\theta}_T\|^2 = \sum_\lambda \Delta_\lambda^2 \, \lambda^{-1} \left[1 - e^{-\lambda T}\right]^2 \tag{14}$$

$$\Delta\mathcal{L}(T) = \sum_\lambda \Delta_\lambda^2 \left[1 - e^{-2\lambda T}\right],$$

where the summation is over all NTK train kernel eigenvalues and $\Delta_\lambda$ is the difference between the network's train outputs at initialization and the target projected on the eigenvector associated with the $\lambda$ eigenvalue.

Making several experimentally motivated assumptions on $\Delta_\lambda$ and $\lambda$, we next derive concrete asymptotic results for the inefficiency ratio. Specifically, we assume $\lambda_k = k^{-\alpha}$ and $\Delta_{\lambda_k}^2 = k^{-\delta}$ where

$k \in 1, \ldots, n$. Assuming $T \propto \lambda_n^{-1}$ such that the lowest mode is partially learned, as well as $\alpha, \delta > 0$, and $0 < \alpha^{-1}(1 - \delta) < 1$ we find the following large $T$ asymptotic

$$\mathcal{W}_2 = \sum_\lambda \frac{\Delta_\lambda^2 [1 - e^{-\lambda T}]^2}{\lambda} \propto T^{\alpha^{-1}(1-\delta)+1}, \tag{15}$$

$$\beta^{-1} R = \sum_\lambda \Delta_\lambda^2 \left[1 - e^{-2\lambda t}\right] \propto T^{\alpha^{-1}(1-\delta)},$$

whereas for $-1 < \alpha^{-1}(1 - \delta) < 0$ we find

$$\beta^{-1} R \propto T^0, \tag{16}$$

and $\mathcal{W}_2$ remains with the same scaling. Remarkably, in the first regime, we find $T_{\text{SL}}(T) \propto T$. Since the proportionality factors are all $O(1)$, this is the optimal behavior in the scaling sense. In contrast, for, $\delta > 1$ we enter the second regime leading to $T_{\text{SL}}(T) \propto T^{1+\alpha^{-1}(1-\delta)}$. The residues in this regime are concentrated in a few particular directions in the eigenspace of the NTK kernel. Noting that the exponent is now smaller than, 1 we obtain a non-optimal behavior in the scaling sense.

Interestingly, if the target is small compared to the outputs at initialization, $\Delta_\lambda^2$ would be dominated by the output of the network at initialization which is given by a random draw from the NNGP. If, based on their similar performance, we ignore differences between the NTK and NNGP spectra, we have that the discrepancy $\Delta_\lambda^2$ scales as $\lambda$. Furthermore, we often observe $\alpha = 1$, placing us exactly at the threshold value between the efficient and inefficient regime.

**Geometric aspects.** Next, we explore some geometrical aspects of the dynamics, namely how different the length $l_\gamma$ of the curve traveled in weight space (see precise definition in Eq. (65)) is compared to the length $l_{\text{geo}} = \sqrt{\mathcal{W}_2}$ of the optimal path, which is a straight line. As shown in Appendix B the lengths of the two curves as a function $T$ scale identically

$$l_{\text{geo}}(T) \propto T^{(\alpha^{-1}+1-\delta/\alpha)/2}, \tag{17}$$

$$l_\gamma(T) \propto T^{(\alpha^{-1}+1-\delta/\alpha)/2}. \tag{18}$$

Interestingly, we find the same asymptotic for the lengths, independent of $\alpha$ and $\delta$ (for $\alpha > 0, \delta > 0$ and $(\alpha^{-1} + 1 - \delta/\alpha) > 0$). This means that at least within this NTK limit, inefficiency is not attributed to having a highly twisted and long curve, but rather having highly inhomogeneous velocity along the curve.

## 4 EXPERIMENTS ON CIFAR-10

Here, we study the efficiency of simple CNNs trained on real-world data. Specifically, we train Myrtle-5, a 5 trainable-layers convolutional network with several pooling layers, having 128 channels on subsets of CIFAR-10 with up to $5k$ samples. Training is carried out for 200k epochs using MSE loss, full batch gradient descent, and small learning rates ($10^{-4}$ to $10^{-5}$) to assure closeness to gradient flow. We train 6 realizations of such networks, with different initialization seeds, and use datasets consisting of the first $n = 500, 1250, 2500, 5000$ samples of CIFAR-10. We record the gradients, losses, and network weights along the path. These enable us to estimate the Wasserstein-2 distance ($L_2$ distance in weight-space) and entropy production (drop in loss) for each realization as a function of time. From these, we obtain the inefficiency ratio per-realization ($T/T_{\text{SL}}(T) = \frac{T \Delta \mathcal{L}(T)}{\|\boldsymbol{\theta}_0 - \boldsymbol{\theta}_T\|^2}$) and geometric inefficiency ratio ($l_\gamma(T)/l_{\text{geo}}(T)$). We furthermore obtain the empirical NTK spectrum (at initialization) and the overlap between initialization residues and the NTK eigenvectors. Notably, while we find these NTK-related quantities useful for a qualitative explanation (see below), this network is not in the strict NTK regime and the kernel does adapt during training (see D.)

As shown in Fig. 1, the very early stages of the dynamics are associated with a fast increase in entropy ($\beta^{-1} R$) or, equivalently, a drop in MSE loss. However, the accuracy does not show any marked features during this process. This motivates us to explore two notions of inefficiency, one measured with respect to the network's initialization (cold start) and the other with respect to the first time at which test accuracy averaged over realizations reached 12% (warm start). We note that for $n = 500$ we reach a final test accuracy of, $30\% \pm 1\%$ whereas for $n = 5000$ we obtain $47\% \pm 1\%$.

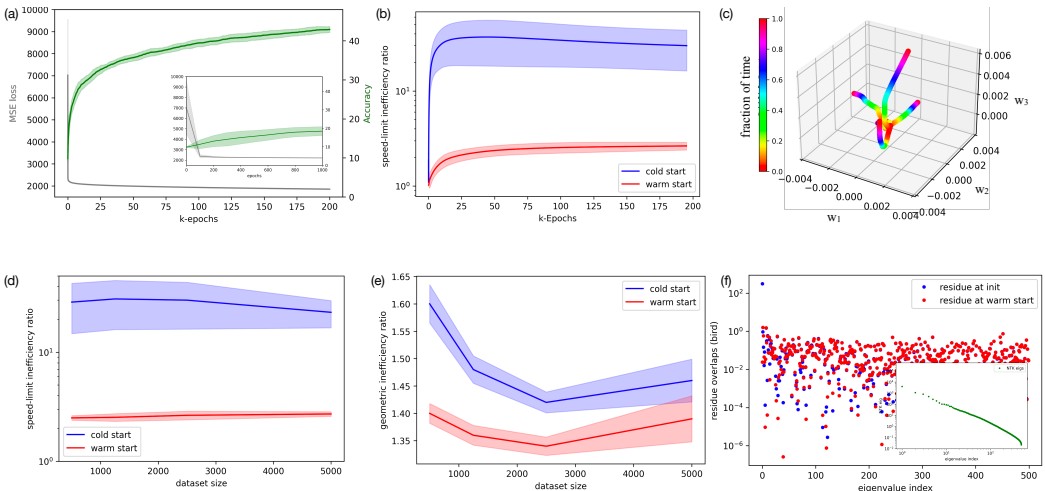

Figure 1: **Efficiency aspects of Myrtle-5 CNN trained on CIFAR-10.** Panel (a): MSE loss and test accuracy for six networks trained on 2500 data points. A dramatic initial decrease in loss is evident without similar improvement in accuracy. Shaded areas reflect standard deviations across networks. Panel (b): For the same networks, speed limit inefficiency as a function of epoch w.r.t. initialization (cold start) or epoch 2000 (warm start). Most of the inefficiency is thus attributed to the fast entropy burn near initialization. Panel (c): Again for $n = 2500$, dynamics of 6 randomly chosen weight-triplets $(w_1, w_2, w_3)$. Panel (d): Inefficiency ratio at epoch 200k as a function of dataset size. Panel (e): Ratio of the curve length traveled during training over the optimal curve, again for different data-set sizes. Panel (f): Overlap of residue at initialization and at the warm start with the NTK eigenvectors. Most of the entropy burn can be associated with the first few eigenvalues, which are quickly learned and hence removed from the residue. Inset: empirical NTK eigenvalues based on a single network with $n = 500$ data points.

Our main results are given in Fig. 1. These support the following rather unexpected picture. Apart from an initial stage at which few very high NTK kernel eigenvalues are learned, the dynamics of this real-world network trained on real-world data seem optimal up to, a roughly constant, $O(1)$ factor. While in principle, one could have expected factors proportional to dataset size or training time, these seem to cancel out. Panel (f) in Fig. 1 complements this picture and can also be explained using our theory. It shows that at initialization the residues are concentrated in a few particular directions in the eigenspace of the NTK kernel. This implies sub-optimal training. At the warm start, in contrast, they are uniformly spread, leading to optimal learning.

Similarly, the length along the curve in weight space traveled during training coincides with the $L_2$ length up to a $O(1)$ factor (panel (e)). This remains true also in the initial, suboptimal phase of training, in line with the theoretical prediction (18), indicating the initial sub-optimality is linked to an increased entropy production by the non-uniform velocity. Panel (c) further tracks several different 3d projections of the path traveled in weight-space (namely the curve $(w_1(t), w_2(t), w_3(t))$ where $w_i$ are some randomly chosen subset of $\boldsymbol{\theta}$) showing rather few twists and turns.

Though the actual NTK kernel of this network is not constant during training, these results are in qualitative agreement with the theoretical results given in the NTK section, where it was assumed constant.

## 5 DISCUSSION

In this work, we set out to explore learning dynamics in deep neural networks from a thermodynamic standpoint. Starting from recent developments in non-equilibrium stochastic thermodynamics on entropy production and thermodynamic speed limits, we developed an analytical framework that makes these concepts available to the deep learning realm. Analytical formulas for these quantities were derived for two simple models, a linear perceptron and a network trained in the NTK regime. Interestingly, following some realistic scaling assumptions on the NTK spectrum, over-parameterized neural networks trained with gradient descent revealed surprising efficiency, leaving only $O(1)$ im-

provement factors to be desired. Similarly, distance-wise, the curve traveled in weight space during training does not differ much from a straight line. Our theoretical results were supported by small-scale experiments on convolutional networks trained in CIFAR-10.

Our speed-limit/time-bound have several potential practical aspects:

**Measurability.** For noiseless training (e.g. SGD at very low learning rates), the Wasserstein-2 distance and entropy production are both readily measurable quantities. For the former is the L2 distance in weight space between the initial weights and the current or final weights. The entropy production is just the difference in whichever loss one is using (MSE/cross-entropy etc...). At weak noise, the entropy production is expressed in terms of the geometry of the loss landscape, namely its gradient and Hessian along the training trajectory. We stress, as reflected in our numerical experiments, that this applies irrespective of whether the final state is in equilibrium or not, and does neither assume infinite width nor a fixed-kernel description. One can measure the time-bound at any time during the dynamics. For noisy training, both quantities are still measurable but as they now involve probability distributions deep-learning-based estimation tools Korotin et al. (2020); Taghvaei & Jalali (2019) can in principle be used. Testing this is left for future work.

**Implications.** As demonstrated in our numerical experiments, measuring $T_{\text{SL}}(T)/T$ provides a dynamical indication of efficiency. It allows one to identify inefficient time segments where improvements are possible and efficient segments where the training is already near optimal. Inefficient segments are characterized by weak entropy production (e.g., a weak decrease of the loss in the noiseless case) accompanied by a strong change in weights. These can be potentially remedied using loss functions that better reflect the change in weights. Alternatively, taking cues from our NTK results, one may engineer the spectral decay of the NTK spectrum and that of the target residues so as to ensure optimality, for example by a suitable initialization. Even in finite-width networks, where the NTK is dynamic, one can sample the adapting kernel during training and use it as a time-local measure of the spectral decay and hence the efficiency.

Various aspects of this work invite further study. It would be interesting to extend our theory to finite learning rates so that it can include discretization effects. This would also shed light on what are the allowed entropy budgets, thereby setting a definite scale for the time-bound. A better quantification of the "warm start" and the time in which optimal training begins is essential. This can also then be connected to the "search" phase and "descent" phase, which were introduced in the context of online learning Bottou & Le Cun (2003); Bottou (2003); Arous et al. (2021); Mandt et al. (2017). The time to escape the search phase was estimated in the context of phase retrieval Arnaboldi et al. (2023). In addition, extending our results to finite-width neutral networks, perhaps using kernel-adaptation methods (Seroussi et al., 2023; Li & Sompolinsky, 2021; Ariosto et al., 2022; Bordelon & Pehlevan, 2023), would enable us to study the thermodynamic implications of feature-learning effects. Another interesting direction would be to study the "warm start" regime, for example by taking an ansatz for the time-dependent change of the kernel, which can be justified in settings as in Atanasov et al. (2021). In this context, it would be interesting to understand how the kernel alignment influences the inefficiency of the learning process, and whether transforming from rich to lazy learning Chizat et al. (2018) affects the inefficiency. Finally, it is desirable to extend our experiments to a wider range of networks, in particular networks and data-sets of larger sizes, and see what practical improvements to training can be gained from this physical viewpoint.

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

# SUPPLEMENTARY MATERIAL

## A  CONDITIONAL DISTRIBUTION, REVERSAL OF TIME AND ENTROPY PRODUCTION

In this section, we provide a derivation of Eq. (23), and Eq. (5) in the main text without loss of generality we take $\eta = 1$. We then show that these two definitions are consistent.

### A.1  ENTROPY PRODUCTION AT EQUILIBRIUM

Similar to, Seifert (2012) we split the definition of the entropy production in Eq. (19) into two parts:

$$R = R_0 + R_1, \tag{19}$$
$$R_0 = \left\langle \ln \frac{p(\boldsymbol{\theta}(0))}{p(\boldsymbol{\theta}(T))} \right\rangle \qquad R_1 = \left\langle \ln \frac{p(\boldsymbol{\theta}(T)|\boldsymbol{\theta}(0))}{q(\boldsymbol{\theta}(0)|\boldsymbol{\theta}(T))} \right\rangle.$$

The initial and final distributions, adopting statistical physics notation, are

$$p_0(\boldsymbol{\theta}) = \mathcal{Z}_0^{-1} \, e^{-\beta \|\boldsymbol{\theta}\|^2} \tag{20}$$

with normalization $\mathcal{Z}_0 = \left(\pi \, \beta^{-1}\right)^{\frac{P}{2}}$ known as the partition function. The temperature determines the variance $(2\beta)^{-1}$ of this Gaussian initial distribution of the weights.

At time $T \to \infty$ the stationary distribution of the weights is

$$p_T(\boldsymbol{\theta}) = \mathcal{Z}_T^{-1} \, e^{-\beta \, (\|\boldsymbol{\theta}\|^2 + \mathcal{L}(\boldsymbol{\theta};\mathcal{D}))}, \tag{21}$$

where $\mathcal{Z}_T = \int e^{-\beta \, (\|\boldsymbol{\theta}\|^2 + \mathcal{L}(\boldsymbol{\theta};\mathcal{D}))} \, d\boldsymbol{\theta}$ is the normalization. The first term, with Eq. (20) and Eq. (21), yields

$$R_0 = \beta \left\langle \|\boldsymbol{\theta}(T)\|^2 + \mathcal{L}(\boldsymbol{\theta}(T);\mathcal{D}) \right\rangle - \beta \left\langle \|\boldsymbol{\theta}(0)\|^2 \right\rangle \tag{22}$$
$$+ \ln \mathcal{Z}_T - \ln \mathcal{Z}_0.$$

The second term $R_1$ measures the log ratio of the process running forward versus backward. For the conservative force in Eq. (1) it can be shown (see Section A.6, Eq. (40)) to take the value

$$R_1 = \beta \left\langle \mathcal{L}(\boldsymbol{\theta}(0);\mathcal{D}) + \|\boldsymbol{\theta}(0)\|^2 \right\rangle - \beta \left\langle \|\boldsymbol{\theta}(T)\|^2 + \mathcal{L}(\boldsymbol{\theta}(T);\mathcal{D}) \right\rangle.$$

So in total with $V$ expressed by Eq. (2) we get the irreversibility

$$\boxed{R = \ln \mathcal{Z}_T - \ln \mathcal{Z}_0 + \beta \left\langle \mathcal{L}(\boldsymbol{\theta}(0)) \right\rangle}. \tag{23}$$

This result expresses the irreversibility of the learning process in terms of equilibrium properties, the free energies of the weight distribution at initialization $\ln \mathcal{Z}_0$ and after learning $\ln \mathcal{Z}_T$ and the expected initial loss.

### A.2  ENTROPY PRODUCTION FROM DYNAMICS

Likewise, entropy production Eq. (19) can be expressed as a dynamical quantity (Vu & Saito, 2022), in terms of the stochastic velocity field $\mathbf{v}(\boldsymbol{\theta}, t)$ (for details see Appendix A.5, i.p. Eq. (38)), which turns the Fokker-Planck equation for the temporal evolution of the density $p(\boldsymbol{\theta}, t)$ into an effective transport equation,

$$\partial_t \, p(\boldsymbol{\theta}, t) + \nabla_{\boldsymbol{\theta}} \cdot \left[ \mathbf{v}(\boldsymbol{\theta}, t) \, p(\boldsymbol{\theta}, t) \right] = 0. \tag{24}$$

Here $\mathbf{v}$ can be thought of as an effective deterministic velocity field that would cause the same evolution of $p(\boldsymbol{\theta}, t)$ as does the stochastic process Eq. (1). Entropy production then takes the form (Appendix A.7)

$$R = \beta \int_0^T \langle \|\mathbf{v}(\boldsymbol{\theta}, t)\|^2 \rangle \, dt, \tag{25}$$

which, in the case of a conservative force of the learning dynamics (cf. Eq. (50)), reads

$$R = \int_0^T \beta \langle \|\nabla_{\boldsymbol{\theta}} V\|^2 \rangle - 2 \langle \Delta_{\boldsymbol{\theta}} V \rangle + \beta^{-1} \langle \|\nabla_{\boldsymbol{\theta}} \ln p\|^2 \rangle \, dt.$$

(26)

In the low noise limit $\beta \gg 1$ the first term $\propto \beta^1$ dominates, which has the simple interpretation of the average squared length of the gradient. The next to this leading term is, $\propto \beta^0$ which contains the average Hessian of the loss function. Equating Eq. (5) and Eq. (23) therefore relates the geometry of the loss landscape to equilibrium properties of the initial and the final distribution of the weights. This is the second theoretical result of this work.

## A.3 Speed limits from optimal transport

The stochastic velocity for $\mathbf{v}$ appearing in Eq. (25) is key to linking entropy production to the distance between the initial and final distribution of the weights, and to the framework of optimal transport. This velocity enables the definition of a measure of the distance between two probability distributions, the Wasserstein-2-distance, (Vu & Saito, 2022, their Eq. (11))

$$\mathcal{W}_2(p_0, p_T) := \min_{\mathbf{v}} T \int_0^T \langle \|\mathbf{v}(\boldsymbol{\theta}, t)\|^2 \rangle \, dt,$$

(27)

where minimization is performed under the constraint that the velocity field $\mathbf{v}(\boldsymbol{\theta}, t)$ for $0 \le t \le T$ transforms $p_0(\boldsymbol{\theta})$ into $p_T(\boldsymbol{\theta})$ by Eq. (24). The right-hand side of Eq. (27) contains the time-averaged mean squared velocity $\langle \|\mathbf{v}(\boldsymbol{\theta}, t)\|^2 \rangle$ required for optimal transport. Comparing Eq. (27) to Eq. (25), the learning process is but one possible transport solution, not necessarily the optimal one though, so one obtains the thermodynamic speed limit known as the Benamou–Brenier formula (Benamou & Brenier, 2000; Vu & Saito, 2022)

$$T \ge \frac{\beta \, \mathcal{W}_2(p_0, p_T)}{R},$$

(28)

which is the third theoretical relation to be explored in the following. It provides a lower bound on the time $T$ for a stochastic process to evolve $p_0$ into $p_T$, which depends on the distance $\mathcal{W}_2$ between the two distributions $p_0$ and $p_T$ and on the amount of entropy $R$ produced at a given temperature $\beta$.

## A.4 Path measure

We here follow (Seifert, 2012, i.p. Sec 4) and (Helias & Dahmen, 2020, i.p. Sec 7.2). Assuming Itô convention, the stochastic differential equation (SDE) Eq. (1) needs to be evaluated in discrete time, as

$$\boldsymbol{\theta}(t + dt) = \boldsymbol{\theta}(t) - \nabla_{\boldsymbol{\theta}} V(\boldsymbol{\theta}(t); \mathcal{D}) \, dt + d\boldsymbol{\mathcal{B}}(t),$$

(29)

$$d\boldsymbol{\mathcal{B}}(t)_i \stackrel{\text{i.i.d.}}{\sim} \mathcal{N}(0, 2\beta^{-1} I_P \, dt).$$

The important point here is that the drift is evaluated at the left boundary $t$ of any time interval $[t, t + dt]$. The dynamics Eq. (1) implies a measure on the path $\boldsymbol{\theta}(t)$ for $t \in [0, T]$. In the following consider discretized time, introducing the temporal indices $l$ as $\boldsymbol{\theta}_l := \boldsymbol{\theta}(l \, dt)$, $d\boldsymbol{\mathcal{B}}_l := d\boldsymbol{\mathcal{B}}(l \, dt)$, and $\mathbf{f}_l := \mathbf{f}(\boldsymbol{\theta}_l, l \, dt) = \nabla_{\boldsymbol{\theta}_l} V(\boldsymbol{\theta}_l; \mathcal{D})$. In this notation, the Ito update step in Eq. (29) takes the form

$$\boldsymbol{\theta}_{l+1} = \boldsymbol{\theta}_l + \mathbf{f}_l \, dt + d\boldsymbol{\mathcal{B}}_l \qquad 0 \le l \le T/dt,$$

(30)

$$\boldsymbol{\theta}_0 = \boldsymbol{\theta}(0).$$

The measure on the path $\boldsymbol{\theta}_1, \ldots, \boldsymbol{\theta}_{T/dt}$ is induced by the Gaussian measure $\propto \exp\big( - \frac{\beta}{4dt} \sum_{i=0}^{T/dt} \|d\boldsymbol{\mathcal{B}}_l\|^2 \big)$ of the stochastic increments $d\boldsymbol{\mathcal{B}}_{l,i} \stackrel{\text{i.i.d.}}{\sim} \mathcal{N}(0, 2\beta^{-1}dt)$. Solving Eq. (30) for $d\boldsymbol{\mathcal{B}}_l = \boldsymbol{\theta}_{l+1} - \boldsymbol{\theta}_l - \mathbf{f}_l \, dt$ one has

$$p(\boldsymbol{\theta}_1, \ldots, \boldsymbol{\theta}_{T/dt} | \boldsymbol{\theta}_0) \propto \exp\big( - \frac{\beta}{4} \sum_{i=0}^{T/dt} \left\| \frac{\boldsymbol{\theta}_{l+1} - \boldsymbol{\theta}_l}{dt} - \mathbf{f}_l \right\|^2 dt \big).$$

(31)

Symbolically, one may therefore write the measure on the path $\boldsymbol{\theta}(0 \leq t \leq T)$ as a functional

$$p[\boldsymbol{\theta}(0 \leq t \leq T)] \propto \exp\big( \int_0^T A[\boldsymbol{\theta}](t)\, dt \big), \tag{32}$$

where $A$ denotes the time-local Lagrangian (also known as the Onsager-Machlup action (Onsager & Machlup, 1953), reviewed in (Helias & Dahmen, 2020, i.p. Sec 7.2))

$$A[\boldsymbol{\theta}](t) = -\frac{\beta}{4}\big[\partial_t \boldsymbol{\theta}(t) - \mathbf{f}(\boldsymbol{\theta}(t), t)\big]^2. \tag{33}$$

Note, however, that in the symbolic notation the Ito procedure as well as the initial condition are both implicit.

### A.5 FOKKER-PLANCK EQUATION AND EQUILIBRIUM DISTRIBUTION

The above process can also be represented in terms of macroscopic quantities, such as probability density. The probability density satisfies the Fokker-Planck equation. This equation takes the form of a continuity equation (cf. Risken (1996))

$$\partial_t\, p(\boldsymbol{\theta}, t) = -\nabla_{\boldsymbol{\theta}} \cdot \mathbf{J}(\boldsymbol{\theta}, t), \tag{34}$$

with the probability current $\mathbf{J}$

$$J(\boldsymbol{\theta}, t) = \big(\mathbf{f}(\boldsymbol{\theta}, t) - \beta^{-1}\, \nabla_{\boldsymbol{\theta}}\big)\, p(\boldsymbol{\theta}, t). \tag{35}$$

For a conservative force $\mathbf{f}(\boldsymbol{\theta}) = -\nabla_{\boldsymbol{\theta}} V(\boldsymbol{\theta})$ the stationary distribution is of Boltzmann form

$$p_0(\boldsymbol{\theta}) \propto e^{-\beta V(\boldsymbol{\theta})}, \tag{36}$$

for which the probability current $J(\boldsymbol{\theta}) \equiv 0$ vanishes. A different way of writing the Fokker-Planck equation Eq. (34) is in the form of a transport equation where the probability current $\mathbf{J} = \mathbf{v}\, p$ is the product of velocity $\mathbf{v}$ and probability $p$, namely

$$\partial_t\, p(\boldsymbol{\theta}, t) = -\nabla_{\boldsymbol{\theta}} \cdot \big[\mathbf{v}(\boldsymbol{\theta}, t)\, p(\boldsymbol{\theta}, t)\big], \tag{37}$$

$$\mathbf{v}(\boldsymbol{\theta}, t) = \mathbf{f}(\boldsymbol{\theta}) - \beta^{-1}\, \nabla_{\boldsymbol{\theta}}\, \ln\, p(\boldsymbol{\theta}, t). \tag{38}$$

The additional term $-\beta^{-1}\, \nabla_{\boldsymbol{\theta}} \ln p$ can be regarded as an entropic force. The interpretation of $\mathbf{v}$ as a velocity makes sense, because it may be interpreted as the probability current $\mathbf{J}$ conditioned on finding the system in state $\boldsymbol{\theta}$ at time $t$. For a system in thermodynamic equilibrium Eq. (36), the velocity vanishes at each point $\boldsymbol{\theta}$, because $\mathbf{v}_0(\boldsymbol{\theta}) = \mathbf{J}_0(\boldsymbol{\theta})/p_0(\boldsymbol{\theta}) = 0$ that is the velocity at $t \to \infty$.

### A.6 IRREVERSIBILITY WITH CONSERVATIVE FORCES

To measure the irreversibility, we need the ratio of probabilities Eq. (19)

$$R_1 = \Big\langle \ln \frac{p(\boldsymbol{\theta}(T)|\boldsymbol{\theta}(0))}{q(\boldsymbol{\theta}(0)|\boldsymbol{\theta}(T))} \Big\rangle, \tag{39}$$

where $p$ denotes the measure Eq. (32) on the path $\boldsymbol{\theta}$ running forward in time and $q$ denotes the probability assigned to a path $\tilde{\boldsymbol{\theta}}$ by the measure Eq. (32) if one reverses the temporal sequence of state traversals

$$\tilde{\boldsymbol{\theta}}(t) := \boldsymbol{\theta}(T - t) \quad 0 \leq t \leq T.$$

The reversed path $\tilde{\boldsymbol{\theta}}$ is constructed such that its initial point $\tilde{\boldsymbol{\theta}}(0)$ is identical to the final point of the forward dynamics $\boldsymbol{\theta}(T)$, so $\tilde{\boldsymbol{\theta}}(0) = \boldsymbol{\theta}(T)$. The average in Eq. (39) is over the ensemble of all paths that started at $t = -\infty$, thus it is identical to the expectation over all random initialization at $t = 0$.

Inserting $\tilde{\boldsymbol{\theta}}$ into the Lagrangian $A$ Eq. (33) only the mixed term $\beta/2\, \mathbf{f}(\boldsymbol{\theta}(t)) \cdot \partial_t \boldsymbol{\theta}(t)$ changes sign, so that Eq. (39) reads

$$\begin{aligned} R_1 &= \beta \Big\langle \int_0^T \big[\mathbf{f}(\boldsymbol{\theta}(t)) \cdot \partial_t\, \boldsymbol{\theta}(t)\big]\, dt \Big\rangle \\ &= \beta \Big\langle \int_{\boldsymbol{\theta}(0)}^{\boldsymbol{\theta}(T)} \mathbf{f}(\boldsymbol{\theta}) \cdot d\boldsymbol{\theta} \Big\rangle \\ &= \beta\, \big(\langle V(\boldsymbol{\theta}(0); \mathcal{D})\rangle - \langle V(\boldsymbol{\theta}(T), \mathcal{D})\rangle\big), \end{aligned} \tag{40}$$

where the penultimate line holds for any non-equilibrium Langevin dynamics with time-independent force $\mathbf{f}(\boldsymbol{\theta}(t))$ and the last line holds in case that $\mathbf{f}$ is conservative. In the latter case, irreversibility depends linearly on the difference in energy $\Delta V$ between initial and final state. Physically, this is the work that the heat bath has exerted on the system (Crooks, 1999, i.p. their Eq. (6)). The irreversibility $R$ defined in Eq. (19) for a conservative force $\mathbf{f} = -\nabla_{\boldsymbol{\theta}} V$ thus is,

$$R = \langle \ln p(\boldsymbol{\theta}(0)) \rangle - \langle \ln p(\boldsymbol{\theta}(T)) \rangle \tag{41}$$
$$+ \beta \left\langle V(\boldsymbol{\theta}(0)) \right\rangle - \beta \left\langle V(\boldsymbol{\theta}(T)) \right\rangle,$$

which corresponds to Eq. (6) in Crooks (1999).

### A.7 IRREVERSIBILITY FROM STOCHASTIC VELOCITY

We here show that, in the case of conservative forces, the irreversibility Eq. (41) obtained from the initial and final equilibrium distribution is identical to the dynamic expression Eq. (25). To show the equivalence, consider the temporal change of the mean of any observable $O(\boldsymbol{\theta})$ is $\partial_t \langle O \rangle = \partial_t \int_{\Omega} p(\boldsymbol{\theta}, t) \, O(\boldsymbol{\theta}) \, d\boldsymbol{\theta}$. Choosing in particular $V$ as the observable $O = V$ the temporal change of the potential is

$$\partial_t \langle V(\boldsymbol{\theta}(t)) \rangle = \int_{\Omega} V(\boldsymbol{\theta}(t)) \, \partial_t \, p(\boldsymbol{\theta}, t) \, d\boldsymbol{\theta} \tag{42}$$
$$\stackrel{(37)}{=} - \int_{\Omega} V(\boldsymbol{\theta}) \, \nabla_{\boldsymbol{\theta}} \cdot \left[ \mathbf{v}(\boldsymbol{\theta}, t) \, p(\boldsymbol{\theta}, t) \right] d\boldsymbol{\theta}$$
$$\stackrel{\text{i.b.p.}}{=} \int_{\Omega} \left[ \nabla_{\boldsymbol{\theta}} V(\boldsymbol{\theta}) \right] \cdot \mathbf{v}(\mathbf{x}, t) \, p(\boldsymbol{\theta}, t) \, d\boldsymbol{\theta}$$
$$= - \int_{\Omega} \mathbf{f}(\boldsymbol{\theta}) \cdot \mathbf{v}(\boldsymbol{\theta}, t) \, p(\boldsymbol{\theta}, t) \, d\boldsymbol{\theta},$$

where we assumed that $p(\boldsymbol{\theta}, t)$ declines sufficiently quickly with $\|\boldsymbol{\theta}\| \to \infty$, so boundary terms vanish when integrating by parts (i.b.p.). So we find

$$\int_{\Omega} \|\mathbf{v}(\boldsymbol{\theta}, t)\|^2 \, p(\boldsymbol{\theta}, t) \, d\boldsymbol{\theta}$$
$$\stackrel{(38)}{=} \int_{\Omega} \left[ \mathbf{f}(\boldsymbol{\theta}) - \beta^{-1} \, \nabla_{\boldsymbol{\theta}} \ln p(\boldsymbol{\theta}, t) \right] \cdot \mathbf{v}(\boldsymbol{\theta}, t) \, p(\boldsymbol{\theta}, t) \, d\boldsymbol{\theta}$$
$$\stackrel{(42)}{=} - \partial_t \langle V(\mathbf{x}(t)) \rangle - \beta^{-1} \int_{\Omega} \left[ \nabla_{\boldsymbol{\theta}} \ln p(\boldsymbol{\theta}, t) \right] \cdot \mathbf{v}(\mathbf{x}, t) \, p(\boldsymbol{\theta}, t) \, d\boldsymbol{\theta}.$$

Integration by parts of the latter integral, again using vanishing boundary terms for $\|\boldsymbol{\theta}\| \to \infty$, it is

$$- \int_{\Omega} \left[ \nabla_{\boldsymbol{\theta}} \ln p(\boldsymbol{\theta}, t) \right] \cdot \mathbf{v}(\boldsymbol{\theta}, t) \, p(\boldsymbol{\theta}, t) \, d\boldsymbol{\theta} \tag{43}$$
$$= \int_{\Omega} \ln p(\boldsymbol{\theta}, t) \, \nabla_{\boldsymbol{\theta}} \cdot \left[ \mathbf{v}(\boldsymbol{\theta}, t) \, p(\boldsymbol{\theta}, t) \right] d\boldsymbol{\theta}$$
$$\stackrel{(37)}{=} - \int_{\Omega} \ln p(\boldsymbol{\theta}, t) \, \partial_t \, p(\boldsymbol{\theta}, t) \, d\boldsymbol{\theta}.$$

The latter integral is identical to

$$- \partial_t \int_{\Omega} \ln p(\boldsymbol{\theta}, t) \, p(\boldsymbol{\theta}, t) \, d\boldsymbol{\theta} \tag{44}$$
$$= - \partial_t \underbrace{\int_{\Omega} p(\boldsymbol{\theta}, t) \, d\boldsymbol{\theta}}_{=0} - \int_{\Omega} \ln p(\boldsymbol{\theta}, t) \, \partial_t \, p(\boldsymbol{\theta}, t) \, d\boldsymbol{\theta}.$$

So together we find the differential form of Eq. (25)

$$\int_{\Omega} \|\mathbf{v}(\boldsymbol{\theta}, t)\|^2 \, p(\boldsymbol{\theta}, t) \, d\boldsymbol{\theta} = - \partial_t \langle V(\boldsymbol{\theta}(t)) \rangle - \beta^{-1} \, \partial_t \int_{\Omega} \ln p(\boldsymbol{\theta}, t) \, p(\boldsymbol{\theta}, t) \, d\boldsymbol{\theta}. \tag{45}$$

Taking the temporal integral over the interval $[0, T]$ we arrive at

$$\beta \int_0^T \int_\Omega ||\mathbf{v}(\boldsymbol{\theta}, t)||^2 \, p(\boldsymbol{\theta}, t) \, d\boldsymbol{\theta} \, dt = \beta \langle V(\boldsymbol{\theta}(0)) \rangle - \beta \langle V(\boldsymbol{\theta}(T)) \rangle \tag{46}$$
$$+ \langle \ln p(\boldsymbol{\theta}(0)) \rangle - \langle \ln p(\boldsymbol{\theta}(T)) \rangle$$
$$\overset{(41)}{=} R.$$

The last line is the difference in the entropy between the initial and final state and the right-hand side is identical to Eq. (41).

The differential form Eq. (45), rewritten more briefly as,

$$\langle ||\mathbf{v}(\boldsymbol{\theta}, t)||^2 \rangle = -\partial_t \left( \langle V(\boldsymbol{\theta}(t)) \rangle + \beta^{-1} \langle \ln p(\boldsymbol{\theta}, t) \rangle \right) \tag{47}$$

has an interesting interpretation. In equilibrium statistical mechanics one has $p(\boldsymbol{\theta}) = \mathcal{Z}^{-1} e^{-\beta V(\boldsymbol{\theta})}$ which, taking the $\ln$ and then the expectation value over $p(\boldsymbol{\theta})$, yields the usual relation

$$F := -\beta^{-1} \ln \mathcal{Z} = \langle V(\boldsymbol{\theta}) \rangle + \beta^{-1} \langle \ln p(\boldsymbol{\theta}) \rangle. \tag{48}$$

between free energy $F$, inner energy $\langle V(\boldsymbol{\theta}) \rangle$, and entropy $S = -k_B \langle \ln p(\boldsymbol{\theta}) \rangle$.

So comparing the right-hand sides Eq. (47) and Eq. (48) and defining a "time-dependent free energy" $F(t) := \langle V(\boldsymbol{\theta}(t)) \rangle + \beta^{-1} \langle \ln p(\boldsymbol{\theta}(t)) \rangle$, one has

$$\partial_t F(t) = -\langle ||\mathbf{v}(\boldsymbol{\theta}, t)||^2 \rangle,$$

which, by the non-negativity of the right-hand side, shows that $F(t)$ is a non-increasing function under the Langevin dynamics. Integrated over time, $t \in [0, T]$ this yields

$$\Delta F = F(T) - F(0) = -\int_0^T \langle ||\mathbf{v}(\boldsymbol{\theta}, t)||^2 \rangle \, dt = -\beta^{-1} R.$$

Using the above formula for $R$ as a function of the velocity field $\mathbf{v}$ and the Fokker-Planck equation for the equilibrium density, $p$,

$$R = \beta \int_0^T \int ||\mathbf{v}(\boldsymbol{\theta}, t)||^2 \, p(\boldsymbol{\theta}, t) \, d\boldsymbol{\theta} \, dt \tag{49}$$
$$= \beta \int_0^T \int ||\frac{\mathbf{J} \, p(\boldsymbol{\theta}, t)}{p(\boldsymbol{\theta}, t)}||^2 \, p(\boldsymbol{\theta}, t) \, d\boldsymbol{\theta} \, dt$$
$$= \beta \int_0^T \int \frac{||\mathbf{J} \, p(\boldsymbol{\theta}, t)||^2}{p(\boldsymbol{\theta}, t)} \, d\boldsymbol{\theta} \, dt$$
$$= \beta \int_0^T \int \frac{||(\mathbf{f}(\boldsymbol{\theta}) - \beta^{-1} \nabla_{\boldsymbol{\theta}}) \, p(\boldsymbol{\theta}, t)||^2}{p(\boldsymbol{\theta}, t)} \, d\boldsymbol{\theta} \, dt$$
$$= \beta \int_0^T \int ||\mathbf{f}(\boldsymbol{\theta})||^2 \, p(\boldsymbol{\theta}, t) - 2\beta^{-1} \, \mathbf{f}(\boldsymbol{\theta}) \cdot \nabla_{\boldsymbol{\theta}} p(\boldsymbol{\theta}, t) + \beta^{-2} \frac{\nabla_{\boldsymbol{\theta}} p(\boldsymbol{\theta}, t) \cdot \nabla_{\boldsymbol{\theta}} p(\boldsymbol{\theta}, t)}{p(\boldsymbol{\theta}, t)} \, d\boldsymbol{\theta} \, dt$$
$$= \int_0^T \beta \langle ||\mathbf{f}(\boldsymbol{\theta}(t))||^2 \rangle + 2\beta^{-1} \langle \nabla_{\boldsymbol{\theta}} \cdot \mathbf{f} \rangle + \beta^{-1} \langle ||\nabla_{\boldsymbol{\theta}} \ln p||^2 \rangle \, dt,$$

we obtain three terms with different powers in $\beta$.

In case of a conservative force, $\mathbf{f} = -\nabla_{\boldsymbol{\theta}} V$ this yields

$$R = \int_0^T \beta \langle ||\nabla_{\boldsymbol{\theta}} V(\boldsymbol{\theta})||^2 \rangle - 2 \langle \Delta_{\boldsymbol{\theta}} V \rangle + \beta^{-1} \langle ||\nabla_{\boldsymbol{\theta}} \ln p||^2 \rangle \, dt, \tag{50}$$

where $\Delta_{\boldsymbol{\theta}}$ is the Laplace operator.

# B   DERIVATION OF NTK-RELATED RESULTS

Consider the gradient flow dynamics of the $i$-th parameter

$$\frac{d\theta_i}{dt} = -\sum_\mu \frac{\partial f_\mu}{\partial \theta_i}(f_\mu - y_\mu), \tag{51}$$

where $f_\mu = f(\boldsymbol{x}_\mu)$ uses NTK parameterization (i.e. weights of order 1 and an explicit $1/\sqrt{\text{width}}$ factor accompanying pre-activations), and $y_\mu$ are the $\mu$-th target for $\mu \in [1, n]$.

Using SVD, we can write

$$\partial_{\theta_i} f_\mu = \sum_{\lambda \in \text{Spec[NTK]}} \sqrt{\lambda} u_{\mu,\lambda} v_{\lambda,i}, \tag{52}$$

where $\lambda$'s are the NTK spectrum and the vectors $u_{\mu,\lambda}$ or $v_{\lambda,i}$ for two different $\lambda$'s are orthogonal. The NTK matrix evaluated at two data points, $\mu, \nu$, is given by $\Theta_{\text{NTK}}(\boldsymbol{x}_\mu, \boldsymbol{x}_\nu) = \sum_{i=1}^P \partial_{\theta_i} f_\mu \partial_{\theta_i} f_\nu$. Multiplying the gradient flow equation with $v_{\lambda,i}$ and summing of $i$ one has

$$\partial_t \theta_\lambda = \sum_i v_{\lambda,i} \frac{d\theta_i}{dt} = -\sum_{i\mu} \sum_{\lambda' \in \text{Spec[NTK]}} \sqrt{\lambda'}(f_\mu - y_\mu) u_{\mu,\lambda'} v_{\lambda,i} v_{\lambda',i} = -\sqrt{\lambda} \Delta_\lambda(t), \tag{53}$$

where $f_\lambda = \sum_\mu f_\mu u_{\mu,\lambda}$ and, similarly with $y_\lambda$ and $\Delta_\lambda(t) \equiv f_\lambda(t) - y_\lambda$. The statement that the NTK does not change with training at infinite width, implies here that the SVD vectors and eigenvalue remain fixed. Furthermore, the original NTK derivation showed that

$$\Delta_\lambda(t) = e^{-\lambda t} \Delta_\lambda(0), \tag{54}$$

plugging this into the last equation we obtain

$$\theta_\lambda(t) = \theta_\lambda(0) + \frac{1}{\sqrt{\lambda}} \left[ e^{-\lambda t} - 1 \right] \Delta_\lambda(0), \tag{55}$$

The Wasserstein-2 distance between a fixed initial state and the state at $t$ simplifies here to the $L_2$ distance, yielding

$$\sum_\lambda (\theta_\lambda(t) - \theta_\lambda(0))^2 = \sum_\lambda \lambda^{-1} \left[ e^{-\lambda t} - 1 \right]^2 \Delta_\lambda(0)^2. \tag{56}$$

Next we note that $\beta^{-1} R$ simplifies here to the decrease in train loss indeed

$$\beta^{-1} R = \eta^2 \int_0^T dt (\nabla V)^2 + O(\beta^{-1}) = -\eta \int_{\boldsymbol{\theta}_0}^{\boldsymbol{\theta}_T} d\boldsymbol{\theta}(\nabla L) + O(\beta^{-1}) \tag{57}$$

$$= -\eta[L(\boldsymbol{\theta}_T) - L(\boldsymbol{\theta}_0)] + O(\beta^{-1}).$$

How optimal are the NTK dynamics? To quantify this, we study the time bound over the actual training time, where the time bound is computed w.r.t. $\theta_\lambda(t)$. The advantage of such a quantity is that it is independent of the arbitrary learning rate (recall we already neglected discretization effects) and hence we can take it to 1. Collecting the above results, this ratio is given by

$$\frac{T_{\text{SL}}(t)}{t} = \frac{1}{t} \frac{\sum_\lambda \lambda^{-1} \left[ e^{-\lambda t} - 1 \right]^2 \Delta_\lambda(0)^2}{\sum_\lambda \frac{\Delta_\lambda(0)^2}{2} - \sum_\lambda \frac{\Delta_\lambda(t)^2}{2}} = \frac{2}{t} \frac{\sum_\lambda \lambda^{-1} \left[ 1 - e^{-\lambda t} \right]^2 \Delta_\lambda(0)^2}{\sum_\lambda \Delta_\lambda(0)^2 \left[ 1 - e^{-2\lambda t} \right]}. \tag{58}$$

For a generic NTK kernel ($\Theta_{\text{NTK}}$) and any finite amount of data the ratio $\frac{T_{\text{SL}}}{t}$ decays as $\frac{2\Delta(0)^T \Theta_{\text{NTK}}^{-1} \Delta(0)}{t|\Delta(0)|^2}$ for large enough $t$. This decay as $1/t$ signifies the fact that from some point onward, only exponentially weak (and hence negligible) learning is taking place.

Less general and more interesting results could be obtained by making some scaling assumptions on $\lambda$ and $\Delta_\lambda(0)$. Specifically, we assume $\lambda_k = \Lambda k^{-\alpha}$ and $\Delta_{\lambda_k}^2(0) = \Delta^2 k^{-\delta}$ where $k \in k_\star, \ldots, n$ for some $k_\star \geq 1$. We further choose $t \approx \lambda_n^{-1}$ such that many modes are learned, but some are still left to be learned.

Consider first the Wasserstein-2 term,

$$\sum_k \frac{\Delta_{\lambda_k}^2(0)[1 - e^{-\lambda_k t}]^2}{\lambda_k} \approx \frac{\Delta^2}{\Lambda} \int_{k_\star}^n dk [1 - e^{-\Lambda k^{-\alpha} t}]^2 k^{-\delta + \alpha}, \tag{59}$$

where our replacement of a summation by an integral is justified for high values of $k$ with an additional finite sum correction that is negligible in the limit of large $n$. As we will show, the contribution from high $k$ diverges with, $T$, and hence these dominate over the low $k$ part of the sum. Next making the substitution $x = \Lambda k^{-\alpha} t$ (or $k = [(\Lambda t)/x]^{\alpha^{-1}}$) we find

$$\frac{\Delta^2}{\Lambda} \int_{\Lambda t n^{-\alpha}}^{\Lambda t} dx k^{\alpha + 1} (\Lambda t)^{-1} [1 - e^{-x}]^2 k^{-\delta + \alpha} = \frac{\Delta^2}{\Lambda} \int_{\Lambda t n^{-\alpha}}^{\Lambda t} dx [(\Lambda t)/x]^{\alpha^{-1} + 2 - \delta/\alpha} (\Lambda t)^{-1} [1 - e^{-x}]^2 \tag{60}$$

$$= \Delta^2 \Lambda^{\alpha^{-1} - \delta/\alpha} t^{\alpha^{-1} + 1 - \delta/\alpha} \int_{\Lambda t n^{-\alpha}}^{\Lambda t} dx x^{-\alpha^{-1} - 2 + \delta/\alpha} [1 - e^{-x}]^2.$$

Noting that $[1 - e^{-x}]^2$ scales as $x^2$ at low $x$ the integral is non-divergent around its lower limit for $\alpha^{-1}(1 - \delta) < 1$, hence taking this lower limit to zero does not change the overall asymptotic. Furthermore, for, $\alpha^{-1}(1 - \delta) + 1 > 0$ the integral is convergent around the upper limit (which is in fact the lower limit of the original $dk$ integration). Hence, as far as the large $t$ asymptotic is concerned, we find

$$\mathcal{W}_2 \to \left[ \Delta^2 \Lambda^{\alpha^{-1} - \delta/\alpha} \int_0^\infty dx x^{-\alpha^{-1} - 2 + \delta/\alpha} [1 - e^{-x}]^2 \right] t^{\alpha^{-1} + 1 - \delta/\alpha}. \tag{61}$$

Next, we apply a similar line of reasoning to $\beta^{-1} R$:

$$\sum_\lambda \Delta_\lambda(0)^2 [1 - e^{-2\lambda t}] = \sum_k \Delta^2 k^{-\delta} [1 - e^{-2\Lambda k^{-\alpha} t}] \approx \int_1^n dk \Delta^2 k^{-\delta} [1 - e^{-2\Lambda k^{-\alpha} t}], \tag{62}$$

using the same substitution of variables we have

$$\int_{\Lambda t n^{-\alpha}}^{\Lambda t} dx k^{\alpha + 1 - \delta} (\Lambda t)^{-1} [1 - e^{-2x}] = \int_{\Lambda t n^{-\alpha}}^{\Lambda t} dx [(\Lambda t)/x]^{\alpha^{-1} + 1 - \delta/\alpha} (\Lambda t)^{-1} [1 - e^{-2x}] \tag{63}$$

$$= (\Lambda t)^{\alpha^{-1}(1-\delta)} \int_{\Lambda t n^{-\alpha}}^{\Lambda t} dx [1/x]^{\alpha^{-1} + 1 - \delta/\alpha} [1 - e^{-2x}].$$

Similarly to the Wasserstein-2 distance, for $\alpha^{-1}(1 - \delta) < 1$ the lower integration boundary is convergent. For $\alpha^{-1}(1 - \delta) > 0$ the top, the integration boundary is also convergent, leaving us with an $t^{\alpha^{-1}(1-\delta)}$ asymptotics. On the other hand, for $\alpha^{-1}(1 - \delta) < 0$ it is divergent and therefore leading to an additional $t^{-\alpha^{-1}(1-\delta)}$. Recalling that $\alpha > 0$ overall we find

$$\beta^{-1} R \to \left[ \Lambda^{\alpha^{-1}(1-\delta)} \int_0^\infty dx [1/x]^{\alpha^{-1} + 1 - \delta/\alpha} [1 - e^{-2x}] \right] t^{\alpha^{-1}(1-\delta)} \quad (1 - \delta) > 0 \tag{64}$$

$$\beta^{-1} R \to \left[ \Lambda^{\alpha^{-1}(1-\delta)} [\alpha^{-1}(\delta - 1)]^{-1} \right] t^0 \quad (1 - \delta) < 0.$$

Collecting these results, one arrives at those of the main text.

### B.1 GEOMETRIC LENGTH OF NTK TRAJECTORY

Next, we address the geometry of the curve in the weight space generated by the training procedure. In general the length of a path $\vec{\gamma}$ parameterized by $\tau$ in Euclidean space is

$$l_\gamma = \int_0^\tau d\tau \sqrt{(\partial_\tau \vec{\gamma})^2} \tag{65}$$

in our NTK context $\tau = t$ and

$$\vec{\gamma}(t) = \left( \frac{1}{\sqrt{\lambda_1}} \left[ e^{-\lambda_1 t} - 1 \right] \Delta_{\lambda_1}(0), \frac{1}{\sqrt{\lambda_2}} \left[ e^{-\lambda_2 t} - 1 \right] \Delta_{\lambda_2}(0), ... \right) \tag{66}$$

where we recall that $\Delta_\lambda(0) = f_\lambda(0) - g_\lambda$ the latter being, respectively, network output and target projected on the $\lambda$ SVD eigenvector. The NTK trajectory length is thus

$$l_\gamma = \int_0^t dt \sqrt{\sum_k \lambda_k e^{-2\lambda_k t} \Delta_{\lambda_k}^2}. \tag{67}$$

To see some explicit dependence on the spectrum, namely that it is power law $\lambda_k = k^{-\alpha}$ and that, $\Delta_{\lambda_k}^2 = \Delta^2 k^{-\beta}$ and hence independent of $k$. Following this, we approximate

$$l_\gamma \approx |\Delta| \int_0^t dt \sqrt{\int_1^d dk k^{-\alpha-\beta} e^{-2k^{-\alpha}t}} = |\Delta| \int_0^t dt \sqrt{\int_{2td^{-\alpha}}^{2t} dx (2t\alpha)^{-1} (x/2t)^{-1/\alpha} (x/2t)^{\beta/\alpha} e^{-x}} \tag{68}$$

$$= |\Delta| \int_0^t dt (2t\alpha)^{-1/2} (2t)^{1/(2\alpha)-\beta/(2\alpha)} \sqrt{\int_{2td^{-\alpha}}^{2t} dx x^{-1/\alpha+\beta/\alpha} e^{-x}},$$

where we used the change of variables $x = 2k^{-\alpha}t$. At least for $\alpha > 1$, the $dx$ integration is non-singular at small $x$ hence for $t \ll 1/\lambda_{\max}$, such that the lowest eigenmodes are non-learnable, we can replace the lower integration boundary by zero. Following this we obtain a lower incomplete gamma function

$$|\Delta| \int_0^t dt (2t\alpha)^{-1/2} t^{1/(2\alpha)-\beta/(2\alpha)} \sqrt{\int_0^{2t} dx x^{-1/\alpha+\beta/\alpha} e^{-x}} \tag{69}$$

$$= |\Delta| \int_0^t dt (2t\alpha)^{-1/2} t^{1/(2\alpha)-\beta/(2\alpha)} \sqrt{\gamma(1-\alpha^{-1}+\beta/\alpha, 2t)}.$$

Notably at large $t$ (and correspondingly large $d$), the above integral is dominated by a $t^{(1+\alpha^{-1}-\beta/\alpha)/2}$ divergence as $\gamma(1-\alpha^{-1}+\beta/\alpha, 2t) \to \Gamma(1-\alpha^{-1}+\beta/\alpha)$. No other factors, outside $O(\alpha)$ or $O(1)$ factor, multiply this divergence. This divergence reflects the fact that the path gets longer as more and more modes are being learned. Examining potential divergences around $t = 0$ (this time for the special case of $\beta = 0$) one can expand around $t = 0$ yielding

$$|\Delta| \int_0^t dt (2t\alpha)^{-1/2} (2t)^{1/(2\alpha)} \sqrt{\gamma(1-\alpha^{-1}, 2t)} \tag{70}$$

$$= |\Delta| \int_0^t dt (2t\alpha)^{-1/2} (2t)^{1/(2\alpha)} \sqrt{(2t)^{1-\alpha^{-1}} \Gamma(1-\alpha^{-1}) e^{-2t} \sum_{j=0}^\infty \frac{(2t)^j}{\Gamma(1-\alpha^{-1}+j+1)}}$$

$$= |\Delta| \sqrt{\Gamma(1-\alpha^{-1})/\alpha} \int_0^t dt (2t)^{-1/(2\alpha)} (2t)^{1/(2\alpha)} e^{-t} \sqrt{\sum_{j=0}^\infty \frac{(2t)^j}{\Gamma(1-\alpha^{-1}+j+1)}},$$

hence for $\alpha > 0$ we see no low $t$ divergence.

These two results, especially the long $t$ divergence, should be compared with the $L_2$ distance of a straight-line trajectory at time $t$ given by

$$l_{\text{geo}}^2 = \sum_\lambda \frac{\Delta_\lambda(0)^2 [1-e^{-\lambda t}]^2}{\lambda} \approx \sum_{k=0}^{t^{\alpha^{-1}}} \frac{\Delta_{\lambda_k}(0)^2}{k^{-\alpha}}, \tag{71}$$

where we made a heuristic approximation and sharply separated learnable and unlearnable modes as those with $t\lambda > 1$ and $t\lambda < 1$ (specifically we took $[1-e^{-\lambda t}]^2$ to be 1 for the former and zero for the latter).

Next, making the same assumptions as those carried for the NTK trajectory, we find

$$l_{\text{geo}} = |\Delta| \sqrt{\int_1^{t^{\alpha^{-1}}} dk k^{\alpha-\beta}} = |\Delta| \sqrt{(\alpha-\beta)^{-1} [t^{(\alpha-\beta+1)/\alpha} - 1]}, \tag{72}$$

thus we find a divergence going as $t^{(\alpha^{-1}+1-\beta/\alpha)/2}$. Comparing both asymptotic we find

$$l_{\text{geo}}(t) \propto t^{(\alpha^{-1}+1-\beta/\alpha)/2} \tag{73}$$
$$l_{\gamma}(t) \propto t^{(\alpha^{-1}+1-\beta/\alpha)/2}.$$

Interestingly, we find the same asymptotic for the lengths, independent of $\alpha$ and $\beta$ (for $\alpha > 0, \beta > 0$ and $(\alpha^{-1} + 1 - \beta/\alpha) > 0$).

## C  LINEAR REGRESSION IN HIGH DIMENSION

Consider the problem of linear regression with scalar output, given a dataset $\mathcal{D}_n = \{\boldsymbol{x}_\mu, y_\mu\}_{\mu=1}^n = \{X, \boldsymbol{y}\}$ where $X \in \mathbb{R}^{d \times n}$, and $\boldsymbol{y} \in \mathbb{R}^n$. Our estimator for the output is a plugin estimator (student model) $\hat{y}(\boldsymbol{x}; \boldsymbol{\theta}) = \boldsymbol{\theta}^{\mathrm{T}} \boldsymbol{x}$. We aim to minimize the loss function $\mathcal{L}(\boldsymbol{\theta}) = \frac{1}{2} \sum_\mu (y_\mu - \hat{y}(\boldsymbol{x}_\mu; \boldsymbol{\theta}))^2$, and find the optimal estimator for $\boldsymbol{\theta}$ via Langevin algorithm with learning rate, $\eta$

$$d\boldsymbol{\theta}(t) = -\eta \left( \sum_{\mu=1}^n (y_\mu - \boldsymbol{\theta}(t)^T \boldsymbol{x}_\mu) \boldsymbol{x}_\mu + c_d \boldsymbol{\theta}(t) \right) dt + \sqrt{2\eta\beta^{-1}} d\boldsymbol{\mathcal{B}}(t). \tag{74}$$

Where we set the weight decay to be $c_d \geq 0$. The equilibrium distribution of this process matches the Bayesian posterior distribution which is independent of the learning rate. To be more concrete, we evaluate the bound given the following noiseless target model $\boldsymbol{y}_\mu = \boldsymbol{\theta}_\star^{\mathrm{T}} \boldsymbol{x}_\mu$ with $\boldsymbol{\theta}_\star \sim \mathcal{N}(0, \alpha/d I_d)$, and $\boldsymbol{x}_\mu$ are i.i.d. vectors with i.i.d. entries independent on $\boldsymbol{\theta}_0$.

In order to calculate the speed limit, we need to evaluate the Wasserstein-2 distance and the entropy production. Due to the linearity of this model, and the Gaussian assumption, all these quantities can be calculated exactly. In particular, both initial and final distributions are Gaussian, i.e., $\boldsymbol{\theta}_0 \sim p_0 = \mathcal{N}(0, (\lambda d)^{-1} I_d)$, and $\boldsymbol{\theta}_T \sim p_T = \mathcal{N}(\boldsymbol{\mu}_T, (\beta)^{-1} \Sigma_T)$, where $\Sigma_T = (XX^{\mathrm{T}} + c_d I_d)^{-1}$. The setting of Eq. (23) requires the weight decay and the variance of the weights at initialization to be the same. In order to make the formalism more general and allow for decoupling between the weight decay and the weight at initialization, we define the weight decay as follows $c_d = cd$, $c = (\lambda + \lambda_2)/\beta$ and $\boldsymbol{\mu}_T = \Sigma_T X \boldsymbol{y}$ where $\lambda_2 \geq -\lambda^{-1}$ in order to keep the weight decay positive. We note that one can also derive Eq. (23) by setting from the beginning different values for weight decay and variance of weights at initialization, and that will amount to the same result.

In the following, we take the learning rate $\eta = 1/n$. This is because we are working with the squared error loss (not the mean squared error loss), so that gradients are proportional to $n$, the number of training points. Hence, for constant $\eta$, the speed of changing weights would naturally scale with, $n$ leading to unstable training for fixed $\eta$ as $n \to \infty$). To avoid this trivial "speedup" (and instability) and instead focus on the interesting scaling, we need to scale the learning rate with the data size. Note that, the Wasserstein-2 is invariant to changes in the learning rate, whereas the $\beta^{-1} R$ will be affected by it. We start by calculating the partition functions, at initialization, $\mathcal{Z}_0 = (2\pi\lambda^{-1}/(d))^{d/2}$, and at the end of the training,

$$\mathcal{Z}_T(\mathcal{D}_n) = \int e^{-\frac{(\lambda+\lambda_2)d}{2}\|\boldsymbol{\theta}\|^2 - \frac{\beta}{2}(\boldsymbol{y}-\boldsymbol{\theta}^{\mathrm{T}}X)^{\mathrm{T}}(\boldsymbol{y}-\boldsymbol{\theta}^{\mathrm{T}}X)} d\boldsymbol{\theta}$$
$$= \left( |\Sigma_T|(2\pi/\beta)^d \right)^{1/2} e^{-\frac{\beta}{2}\|\boldsymbol{y}\|^2 + \frac{1}{2}\beta \boldsymbol{y}^{\mathrm{T}}X^{\mathrm{T}}\Sigma_T X \boldsymbol{y}}. \tag{75}$$

The entropy production (Eq. (23)) is then,

$$(n\beta)^{-1}R = (n\beta)^{-1}\log\mathcal{Z}_T(\mathcal{D}_n) - (n\beta)^{-1}\log\mathcal{Z}_0 + \frac{1}{n}\langle\mathcal{L}(\boldsymbol{\theta}(0))\rangle \tag{76}$$

$$= \frac{\gamma_n}{2\beta}\log(d(\beta/\lambda)^{-1}) + \frac{1}{2\beta n}\log|\Sigma_T| + \frac{1}{2n}\|\Sigma_T^{-1/2}\boldsymbol{\mu}_T\|^2 + \frac{1}{2\lambda dn}\text{Tr}\left(XX^{\mathrm{T}}\right)$$

$$= \frac{\gamma_n}{2\beta}\log(d\lambda(\beta)^{-1}) - \frac{1}{2n\beta}\log|cI_d + \frac{1}{d}XX^{\mathrm{T}}|$$

$$+ \frac{1}{2d^2 n}\text{Tr}\left(\left(cI_d + \frac{1}{d}XX^{\mathrm{T}}\right)^{-1}\left(XX^{\mathrm{T}}\right)^2 \boldsymbol{\theta}_\star\boldsymbol{\theta}_\star^{\mathrm{T}}\right) + \frac{1}{2\lambda dn}\text{Tr}\left(XX^{\mathrm{T}}\right), \tag{77}$$

where $\gamma_n = d/n$. In the special case that the initial weights are deterministic with $\boldsymbol{\theta}_0 = 0$, and $\lambda \to \infty$, the loss at initialization is equal to $\frac{1}{2}\|\boldsymbol{y}\|^2$. The entropy production is then reduced to the following formula,

$$\beta^{-1}R = \frac{1}{2d^2 n}\text{Tr}\left(\left(cI_d + \frac{1}{d}XX^{\text{T}}\right)^{-1}\left(XX^{\text{T}}\right)^2\boldsymbol{\theta}_\star\boldsymbol{\theta}_\star^{\text{T}}\right), \tag{78}$$

where $c = \lambda/\beta$ now. Therefore, the dominant contribution is mainly from the bias in our model. Next, since both distributions at initialization and at the end of training are Gaussian, the Wasserstein distance can be calculated exactly,

$$W^2(p_0, p_T) = \|\mu_0 - \mu_T\|^2 + \text{Tr}\left(\Sigma_0 + \beta^{-1}\Sigma_T - 2\beta^{-1/2}\left(\Sigma_T^{1/2}\Sigma_0\Sigma_T^{1/2}\right)^{1/2}\right)$$

$$= \|\mu_T\|^2 + \lambda^{-1} + \beta^{-1}\text{Tr}\left(\Sigma_T\right) - 2(\beta/\lambda)^{-1/2}d^{-1/2}\text{Tr}\left(\Sigma_T^{1/2}\right)$$

$$= \frac{1}{d^2}\text{Tr}\left(\left(cI_d + \frac{1}{d}XX^{\text{T}}\right)^{-2}\left(XX^{\text{T}}\right)^2\boldsymbol{\theta}_\star\boldsymbol{\theta}_\star^{\text{T}}\right) + \lambda^{-1} + \frac{1}{\beta d}\text{Tr}\left(\left(cI_d + \frac{1}{d}XX^{\text{T}}\right)^{-1}\right)$$

$$- 2(\beta/\lambda)^{-1/2}d^{-1}\text{Tr}\left(\left(cI_d + \frac{1}{d}XX^{\text{T}}\right)^{-1/2}\right),$$

where $\Sigma_0 = 1/(\lambda d)I_d$ The speed limit bound, Eq. (28), is then,

$$T(\mathcal{D}_n) \geq \frac{W^2(p_0, p_T)}{\eta\beta^{-1}R} \equiv T_{\text{SL}}. \tag{79}$$

We note that the analysis here can be generalized to other data distributions (see section B). In the regime, where $\gamma_n = d/n \to \gamma \in (0, \infty)$, and $d, n \to \infty$, the results simplify. Taking expectation over $\boldsymbol{\theta}_\star$ and using the concentration of quadratic forms (that can be made rigorous by using the Hanson-Wright inequality Rudelson & Vershynin (2013)), the speed limit bound is, then,

$$T_{\text{SL}} = \frac{\lambda^{-1} + \alpha\int(c\gamma + s)^{-2}s^2 d\rho(s) + \beta^{-1}\int(c + s/\gamma)^{-1}d\rho(s)}{-\frac{\gamma}{2\beta}\log(\lambda^{-1}\beta) - \frac{\gamma}{2\beta}\int\log|c + s/\gamma|d\rho(s) + \frac{\lambda^{-1}}{2}\int sd\rho(s)}$$

$$- \frac{2(\beta/\lambda)^{-1/2}\int(c + s/\gamma)^{-1/2}d\rho(s)}{-\frac{\gamma}{2\beta}\log(\lambda^{-1}\beta) - \frac{\gamma}{2\beta}\int\log|c + s/\gamma|d\rho(s) + \frac{\lambda^{-1}}{2}\int sd\rho(s)} + o(1), \quad (80)$$

where $\rho$ here is the limiting measure of the eigenvalues of $\frac{1}{n}XX^T$ for i.i.d entries and samples, known as the Marchenko–Pastur distribution. It takes the following form

$$\rho(x) = \begin{cases} (1 - \frac{1}{\gamma})\delta(x) + \nu(x), & \text{if } \gamma > 1 \\ \nu(x), & \text{if } 0 \leq \gamma \leq 1, \end{cases}$$

with, $\nu(x) = \frac{1}{2\pi}\frac{\sqrt{(\gamma_+ - x)(x - \gamma_-)}}{\gamma x}\mathbf{1}_{x\in[\gamma_-, \gamma_+]}$, such that, $\gamma_\pm = (1 \pm \sqrt{\gamma})^2$ where $\delta(x)$ is the Dirac delta function.

Interestingly, taking the limit of $\beta \to \infty$ in Eq. (80) (note that $c \to 0$, because $c = (\lambda_2 + \lambda)/\beta$). The speed limit is then

$$T_{\text{SL}}(\beta \to \infty) \to \frac{\lambda^{-1} + \alpha}{\frac{\lambda^{-1}}{2}\int sd\rho(s)}. \tag{81}$$

Note that, taking now the limit of $d \to \infty$ ($\gamma \to \infty$) corresponds to $d \gg n$, we get that

$$\lim_{d\to\infty}\lim_{\beta\to\infty}T_{\text{SL}} = 2(1 + \alpha\lambda).$$

In this regime, the weight decay term generates additional noise due to over-parametrization.

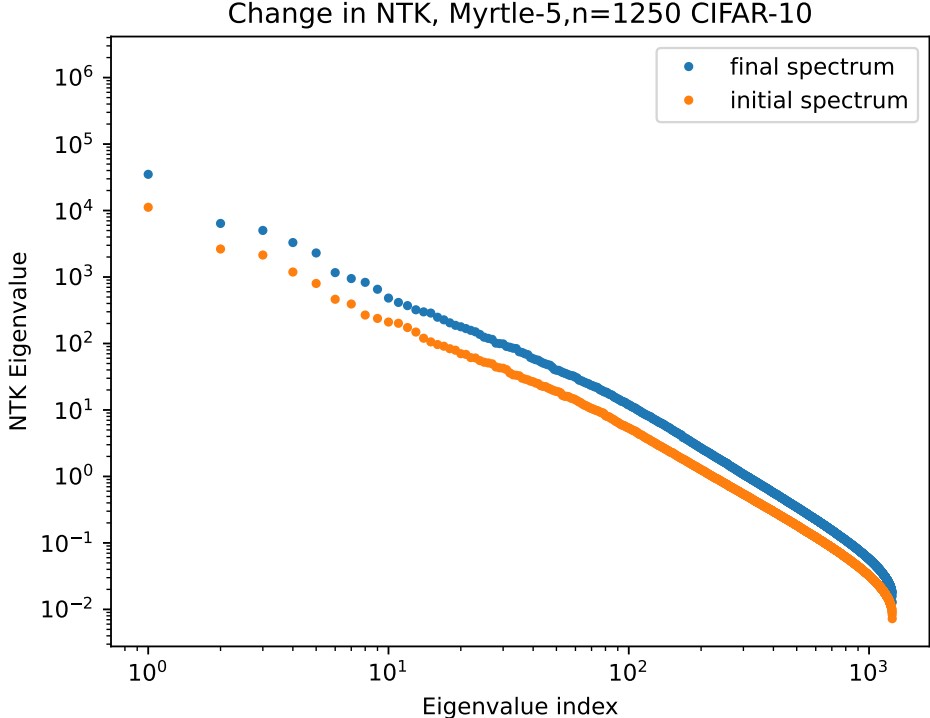

Figure 2: Empirical NTK spectrum obtained at initialization and after 200k epochs, for the myrtle-5 experiments cited in the main text with 1250 data points. An order 1 difference in the spectra which decay towards the lowest eigenvalues shows that one is not close to the strict NTK regime.

On the other hand, if we take $\beta \to 0$ we get

$$T_{\text{SL}}(\beta \to 0) \to 0. \tag{82}$$

In this regime, the system is driven by noise, and there is essentially no learning. In the over-parametrized regime in which $d \to \infty$, and $d \gg n$ ($\gamma \to \infty$) we have that

$$T_{\text{SL}}(d \to \infty) \to 0. \tag{83}$$

This shows that when the system is extremely over parametrized the distribution is barely moving from its initial condition. We note that as shown above that will not be the case in zero noise. I.e. the limit of $d \to \infty$ does not commute with the limit of $\beta \to \infty$.

Last, as $n \to \infty$, ($\gamma \to 0$) corresponds to $n \gg d$, the bound reaches the following finite value:

$$T_{\text{SL}}(n \to \infty) \to 2\alpha\lambda. \tag{84}$$

This limit is in essence where we learn the population error i.e. the expectation of the loss function over the true dataset distribution. Remarkably, it is independent of the amount of noise $\beta$.

## D CHANGE IN KERNEL DURING MYRTLE-5 EXPERIMENTS.

In the reported numerical experiments, Myrtle-5 network with $C = 128$ channels was trained on a few thousand CIFAR-10 images. Both in theory and in practice, (Seroussi et al., 2023; Lee et al., 2019; Jacot et al., 2018), convergence to the NTK regime is obtained when $C$ is taken to infinity at finite $n$, where often $C \gg n$ suffices. Indeed, in our case, taking $n = 1250$ we find that empirical NTK spectrum changes by factors of order one, between the initial and final state of the network as demonstrated in Fig. 2. We emphasize, though, that our measurement of $T_{\text{SL}}(t)$ does not make any NTK assumption, apart from approximately continuous and noiseless dynamics.

