# OpenReview forum: "Speed Limits for Deep Learning"
_ICLR.cc/2024/Conference — Submitted to ICLR 2024_

### Official Review · Reviewer_5tEv · 2023-10-27

**Soundness:** 3 good
**Presentation:** 3 good
**Contribution:** 2 fair
**Rating:** 6
**Confidence:** 2

**Summary:**

This problem considers the continuous dynamic of gradient-based learning problems. Studying the problem from a thermodynamic perspective, the paper derives a lower bound on the time for a process to go from an initial state to a target state using Wasserstein-2 distance. The paper further considers two realizations of the problem, namely the linear regression and the NTK, and presents the implication of the result under various limiting scenarios.

**Strengths:**

1. It seems an interesting idea to connect the machine learning optimization dynamic with notions in thermodynamics.

2. The paper provides interesting interpretations of the speed limit in linear regression and NTK learning.

**Weaknesses:**

1. I am not sure about the significance of this theoretical investigation. It seems that the paper only considers the lower bound of the time that goes from an initial parameter state to a target parameter state. It does not tell us information about e.g. the time lower bound to get to a near-stable parameter state, or the time lower bound to get to near-zero potential.

2. Characterizing the speed limit using the Wasserstein-2 distance is not easily interpretable since it is in general hard to compute the Wasserstein-2 distance. The paper seems only able to derive interpretable results under limiting conditions like no noise or infinite parameters.

3. The models considered in this paper are fairly simple. Both the linear regression and the learning under NTK assumption are linear models and are not representative of deep learning in general.

4. The writing of the paper can be improved. I hope to see formal theorems in the paper stating the main contribution, and notations need to be clearly stated. For instance, I am not sure what is $Z_T$ and $Z_0$ in Eq. (4). There is also no related works section.

**Questions:**

Given the conclusion in the paper that in the NTK regime the dynamic in the paper achieves a rate that is almost optimal, is this contradicting the fact that the continuous heavy-ball dynamic achieves a faster rate to converge to the stable point (see, e.g. https://link.springer.com/article/10.1007/s1022-01164-w)?

---

> ### Author Response · Authors · 2023-11-16
> **Thank you very much for your review. We would first like to comment that the revised version contains a more detailed discussion of practical implications in the discussion section.**
>
> Reply to weakness:
>
> 1. The time bound is interesting because it sets a threshold of optimal learning. Unlike what the referee suggests, the speed limit bounds in the paper hold quite generally whether one is in an equilibrium state or not (see for instance Eq. 8 in the revised manuscript). Admittedly, computing it analytically during the dynamics and finite-width networks may be challenging. However, in our NTK section, we compute it analytically for infinite-width networks. Furthermore, in our numerical experiment, we compute it numerically for a finite-channel CNN working well outside the NTK regime and still find that the results are in qualitative agreement with NTK analytics (see also new App. Sec. D.)
>
> 2. Based on your feedback, we've extended the discussion at the end of section 2.2 and the discussion section to emphasize that besides the analytical investigations like those we undertake in the paper, one could hope to study much more realistic scenarios (including ones with SGD noise) by numerically bounding the Wasserstein distance (using techniques like those in "2-Wasserstein Approximation via Restricted Convex Potentials with Application to Improved Training for GANs" by Taghvaei and Jalali, or "Wasserstein-2 Generative Networks" by Korotin et al.; references we've added) and by numerically bounding or calculating the entropy production directly, closer to Section 4. The latter is predominantly determined by the geometry of the loss landscape, in terms of the gradient and the Hessian along the training trajectory (Eq (5)). With help of contemporary finite-size network theory (e.g., Li \&,Sompolinsky 2020, Hanin\&Slopaka 2022, Zavatone-Veth\&Pehlevan (2021), Seroussi\&Ringel 2023), moreover, analytical estimates of the partition function required by our Eq. (4) for the entropy production, are within reach.
> Moreover, since all quantities involved in the time bound are readily measurable (see new paragraph "measurability" in the discussion), the inefficiency ratio may be used as an indicator to identify periods within the training that are furthest from optimality and hence would predominantly benefit from optimization, thus serving as a guide for the development of training algorithms.
> Finally, we comment that even using finite-batch SGD our bound still applies in the limit of very low learning rates, as the SGD noise averages out giving us again gradient flow type behavior. It seems plausible to us, that improving the architecture/training hyperparameters so that the dynamics travels closer to the time bound in this low learning rate region, would also improve the behavior at higher learning rates used in experiments.
>
> 3. We would first like to point out that a major part of this work is really about bridging the two worlds of stochastic thermodynamics and deep learning. For instance, our expressions for entropy production due to learning [Eq. (4,5,6) in the revised version] are, to the best of our knowledge, new results. In the final version, we plan to highlight those results as theorems for which this status can be justified from a mathematical point of view.
> Still, as you point out, in this work we consider fairly simple models: high-dimensional linear regression, the NTK, but also a more realistic finite-channel Mytrle-5 network trained on cifar10. We felt that these fairly simple scenarios, besides being more analytically tractable, were easier to interpret and reason about.
> Notwithstanding, there is a clear roadmap to extend these results to networks in the feature learning regime using recent DMFT results or adaptive kernel approaches wherein feature learning is reflected by an augmented GP kernel governing the outputs layer and the hidden layers. For instance, using the approaches of https://www.nature.com/articles/s41467-023-36361-y one can readily estimate $\log(Z_{T})$ at $T \rightarrow \infty$ and using the approach https://arxiv.org/abs/2111.00034 similarly calculate the time bound. As said, this paper is mainly about introducing this new concept and its potential applications. One of the reasons we find it exciting is that it opens the door to a fundamental understanding of the efficiency of learning with ample future possibilities.
>
> 4. Thank you for pointing out this notation issue, which was indeed unclear. This has been fixed in the revised version. We added a subsection in the introduction describing related work, as well as related literature in the discussion.

---

> > ### Author Response · Authors · 2023-11-16
> > **Reply to Questions:**
> >
> > Regarding the final question, the included link doesn't seem to work, so we're not sure which reference you meant, could you clarify?  In general, though, different dynamics will have different entropy production rates.  It would be interesting, and we hope to encourage others to measure entropy production rates for different optimization schemes.  It seems likely that differences in observed convergence times equate to measurable differences in the entropy production for different methods. Alternatively, we note that the notion of optimality we found in the NTK section is meant in the scaling sense - namely that $T_{SL}(t)/t$ does neither scale up with the number of data points nor with the training time. This still leaves an $O(1)$ room for improvement.

---

> > > ### Comment · Reviewer_5tEv · 2023-11-20
> > > **Further clarification of the questions.**
> > >
> > > I am referring to the paper: "Convergence rates for the Heavy-Ball continuous dynamics for non-convex optimization, under Polyak-Łojasiewicz condition" (hopefully this link will work: https://arxiv.org/abs/2107.10123). It seems that the heavy ball dynamic indeed has an acceleration effect in some scenarios. Does this contradict the argument in the paper that the GD achieves an optimal speed limit (if we ignore the constant)?

---

> > > > ### Author Response · Authors · 2023-11-22
> > > > **Concerning Heavy Ball dynamics**
> > > >
> > > > Thank you for this very interesting question.
> > > >
> > > > This could be an additional application of the speed limit bound. Our speed limit is currently with respect to Gradient descent (overdamped Langevin dynamics) without momentum term. However, one could add a momentum term to our analysis by considering a system of equations both for the parameter and its derivative ("position" and "momentum"). Taking the relevant case of similar momentum distribution at initialization and at the final state, we expect that W_2 would remain the same (this is evidently the case in the noiseless case). The interesting part comes from the calculation of the entropy production which becomes a bit more intricate. In particular, one would need to include the possibility of anisotropic noise (one for the position part and the momentum part), since in general, it makes sense to take non-zero noise in the momentum direction and zero noise in the position. The equation for the entropy production can be derived in this case by introducing the path measure which will contain now two components both for the position part and the momentum part (See Appendix A.4). Then following the derivation in Appendix A.6 and A.7 arrive to a modified equation for R.
> > > >
> > > > Intuitively, we expect the entropy production to increase since adding momentum adds more degrees of freedom, and introduces larger irreversibility in time due to the acceleration. Therefore, the speed bound would decrease. Leading to a more optimal result. We would add a discussion about it in the final version of the paper including the revised equation for the entropy production.

---

> > ### Comment · Reviewer_5tEv · 2023-11-20
> > **Response to the author's rebuttal**
> >
> > Thank you for your effort in updating the paper and clarifying my concerns. Indeed, I agree that this direction is only in its starting phase and what we can do right now is quite limited. While my concerns remain about whether this analysis can be extended to more realistic settings, given that my other questions are resolved and this is a fairly novel paper, I would like to increase the rating from 5 to 6.

---

### Official Review · Reviewer_M1Mw · 2023-10-31

**Soundness:** 3 good
**Presentation:** 3 good
**Contribution:** 3 good
**Rating:** 8
**Confidence:** 4

**Summary:**

The paper applies recent advances in stochastic thermodynamics to analyze the efficiency of training neural networks. It derives analytical expressions relating the speed limit (minimum training time) to the Wasserstein 2-distance between initial and final weight distributions and the entropy production. For linear regression and neural networks in the NTK regime, exact formulas are provided for the quantities involved in the speed limit. Under plausible assumptions on the NTK spectrum (power law behavior) and residue (defined as the target minus the initial prediction), NTKs exhibit near-optimal training efficiency in the scaling sense. Small-scale experiments on CIFAR-10 qualitatively support the theoretical findings.

**Strengths:**

I think this is a technically sound paper that makes good contributions. The application of stochastic thermodynamics concepts to neural network training is novel. I have not seen it before in prior literature. The analysis done in the paper is insightful and to the best of my knowledge seems mathematically rigorous. The results on optimal scaling efficiency are intriguing. The writing is clear and relatively compact. It seems like the relevant prior works cited correctly. Moreover this paper provides a good literature review on various different topics across entropy production and the various bounds on that quantity.

**Weaknesses:**

It is unclear if the near optimal scaling efficiency result applies to large realistic models and datasets. The CIFAR-10 study used very small networks. It would be very nice to see an empirical example with a larger scope.

Further, it would be nice to be more explicit of how much of the entropy production is due to the presence of nonzero initial weights. It would be nice to cover the case of either the perceptron or the NTK starting with $\theta_0 = 0$.

It is rather unclear whether there are any takeaways from practitioners. Its not strictly necessary that their should be, but given the title one is left to wonder whether there are possible statements that can be made about the compute-optimal frontier.

**Questions:**

It's interesting that power law scalings in the target (ie in the residues) seem to imply an inefficiency factor that grows with dataset size. Can the authors comment on whether this is representative of realistic datasets?

The initial transient period seems important. The current characterization is that the low modes are learned very quickly during this period, however many other things are also happening. For one, the kernel could be drastically realigning its eigenstructure (as in e.g. Atanasov Bordelon Pehlevan https://arxiv.org/abs/2111.00034). Relatedly, it would be interesting to see these empirical results for the NN as the feature learning parameter (as in $\alpha$ in Chizat and Bach https://arxiv.org/abs/1812.07956) is varied.

It seems strange that in the high noise regime the formalism tells the perceptron learns in "zero time" when really its unable to learn at all. Am I understanding this correctly?

To the best of my understanding, the only setting in which the W2 distance enters practically is when the marginals are delta functions. So it only is realized as the 2-norm in weight space. Is there any use to the optimal transport formalism beyond this?

---

> ### Author Response · Authors · 2023-11-16
> **We thank the reviewer for this concise summary and for the insightful and helpful comments we received from their review.**
>
> Weakness:
>
> 1. We completely agree with you. This work however is more theoretical in nature and shows the potential of the speed bound as an object of study. We plan to investigate numerically the speed limit in various settings, performing large-scale experiments in follow-up works. To pick up this point, we have amended the outlook section of the discussion in this direction.
>
> 2. Thank you for your question, this is an interesting limit. We added an analysis of this case in the linear perceptron in Appendix C of the paper. In this case, the initial loss is just the norm of the target. Calculation of the final and initial free energies shows that the norm of the target is canceled by a similar term in the free energy, and the term with the significant contribution to the entropy production is the bias of the linear regression estimator.
>
> 3. This is an important point. We agree that our paper in its current form focuses on theoretical insights, provided by a direct computation of the speed bound. The analysis indicates that the optimality is tightly connected to the spectrum of the NTK and the projection of the residuals. One important message is that when the residuals are uniform, the learning process is optimal. The assumed power law distributions are, moreover, observed in various applied settings (see, e.g., Moloney et al. 2022).
> An additional practical use of the speed limit is as an additional quantity to calculate, which gives an indication of whether the optimization and the training process are efficient.  In fact, for gradient descent with infinitesimal step size ($\beta \to \infty$), we can calculate the speed limit numerically very efficiently since it is the Euclidean distance between the initial and final weights divided by the difference in the loss function, i.e. $\frac{\|\theta_0-\theta_T\|^2}{\mathcal{L}(\theta_0)- \mathcal{L}(\theta_T)}$. For small noise, the denominator can be computed from the slope and Hessian of the loss along the training trajectory. For a general noise level, you are right that this would be harder. However, there is recent research suggesting ways of calculating the W2 distance efficiently. We added a discussion about that in the text.
> Another practical insight is that the inefficiency of training is mostly
> tied to the initial period of the training process. Concretely, the
> link of inefficiency to the shape of the power law of residuals at
> initialization opens the door to design specific weight initialization
> that bring this exponent into the optimal regime.

---

> > ### Author Response · Authors · 2023-11-16
> > **Reply to questions:**
> >
> > 1. Thank you for this helpful question. We note that our initial presentation has not been clear enough in this respect. In fact, the inefficiency ratio does not grow with the dataset size. This is also apparent in Figure 1(d) which shows that the inefficiency ratio does not depend on the dataset size. Our main observation is that if the residue projections on the kernel eigenvalue are too concentrated in a few directions, then this will increase the inefficiency. We added clarification of this important point in the main text in the NTK section 3.2 and in the experimental results section 4.
> >
> > 2. This is indeed counterintuitive. In this limit, the initial and final distributions are the same and therefore the bound predicts the learning to happen at zero time. In this sense, one must remember that this is a lower bound and that it does not tell us something about the ability of the estimator to generalize. We added a clarification about this in the linear regression section.
> >
> > 3. Thank you for your suggestion. This is an interesting direction to explore. Our current theoretical analysis is limited to the infinite width case in which the kernel eigenfunction does not change in time. It would indeed be interesting to extend this idea to the case of the time-dependent case and finite width limit, considering similar ansatz as in Eq. (3) in Atanasov Bordelon Pehlevan (2021) https://arxiv.org/abs/2111.00034. Indeed, this would allow us to explore the inefficiency ratio in the mean-field scaling, in which the weights are very relevant. Our current conjecture is that though the W2 distance may explode, it would be moderated by the learning rate which will also be scaled accordingly, yielding in the end a constant time-bound. We added a discussion about that in the discussion section.
> >
> > 4. The derivation of the speed limit formula relies on ideas rooted in optimal transport. The formula by Benamou–Brenier 2000 for W2 distance provides another way of calculating the W2 distance (the optimal transport plan) using the continuity equation in fluid dynamics. The speed limit is derived by noting that the entropy production is a transport plan, not necessarily the optimal one. In addition, we are also using the W2 distance between Gaussian when calculating the bound for linear regression. This is explained in the speed limit from optimal transport section 2.2, we also add clarification that in the linear regression setting, since the initial and final distribution are Gaussian distributions, one can calculate the W2 and the entropy production for general noise level $\beta$.

---

> ### Comment · Reviewer_M1Mw · 2023-11-21
> **Response to Authors' Rebuttal**
>
> I thank the authors for their thoughtful rebuttal and clarifications on my questions about the paper. I hope that the authors will add these clarifications and additional discussion about connections to more realistic settings. I think that the paper is quite thorough and novel and I recommend acceptance. I've raised my score appropriately.

---

### Official Review · Reviewer_8rXj · 2023-11-06

**Soundness:** 3 good
**Presentation:** 3 good
**Contribution:** 2 fair
**Rating:** 6
**Confidence:** 2

**Summary:**

This paper proposes a study of the "time" it takes for a neural network (NN) to travel from its initialization distribution to its final distribution (after training). The study is based on of the transport of the distribution of the parameters and the related evolution of the entropy of the system. To use this theoretical framework, it is necessary to do some assumptions: continuous-time training, full-batch optimization, simplified models (linear regression, Neural Tangent Kernel (NTK) setting), etc.

This theoretical study comes with a series of experiments, with a setting as close as possible as the theoretical assumptions (small learning rates, full-batch gradient descent). The experimental results are not entirely consistent with the theoretical predictions. The authors claim that the "training time" they have computed theoretically is close to the actual training time of NNs in the experiments.

**Strengths:**

## Originality

To my knowledge, this work is original. But I am not a specialist of statistical physics applied to NNs.

## Clarity

The authors made the effort to make their paper understandable to the reader who would not be a specialist in statistical physics applied to NNs. Overall, the paper is easy to read.

## Quality

The experimental section, despite being narrow (only one setting has been tested), provides enough results to evaluate the significance and the limitation of the theoretical section.

**Weaknesses:**

## Significance

### Narrowness of the theoretical setting

Only two setups have been studied: linear regression and NNs in the NTK regime. Moreover, the continuous-time SGD does not model faithfully the discrete SGD when training practical NNs on realistic data.

Moreover, the authors does not discuss how $T_{SL}$ (lower bound on the training time) obtained in the NTK regime compare to a hypothetical $T_{SL}$ obtained in finite-width NNs. Would it be larger of smaller? ...

### Motivation

Given the theoretical framework, I do not fully understand how this work is related to the usual challenges in deep learning. Can we use this work to improve optimization? to obtain theoretical guarantees? ...

**Questions:**

Main questions:
 * motivation: can we use this work to evaluate the quality of an optimizer?
 * stronger results: how does the $T_{SL}$ obtained in the NTK limit relate to some "$T_{SL}$" in the finite width setting?
 * experimental setup: the authors claim that a learning rate of $10^{-5}$ is small and apply the NTK setting to Myrtle-5; how to justify these choices? Is $10^{-5}$ really small enough? is Myrtle-5 wide enough to consider that we are in the NTK regime?

Other questions:
 * Eqn (10) is difficult to interpret: in the proof, the effective learning rate is $\eta/n$; how such a proof can be interpreted in the limit $n \rightarrow \infty$?
 * I may have misunderstood one point: Figure 1.b seems to indicate that the training time is about $10$-$30$ times larger than $T_{SL}$. We are far from the "$O(1)$" factor written in the claims... while it is true that the ratio of the trajectory lengths (Fig. 1.e) is of order 1. How to solve this contradiction?
 * it is clear to me that the result written in Eqn. (10) depends on the distribution of the data. To obtain such a result, the data are assumed to be Gaussian. It is then not surprising that Fig. 1.d contradicts Eqn. (10), since CIFAR-10 images are far from being Gaussian vectors.

---

> ### Author Response · Authors · 2023-11-16
> **We thank the reviewer for this concise summary and for the insightful and helpful comments we received from their review. These have considerably helped us to clarify the message.**
>
> Reply to Weakness:
>
> Significance reply by points:
>
> Point 1:
>
> This is an important insight, and we are grateful to the reviewer for
> pointing it out. We agree that usual training algorithms employ
> discrete updates for training, and we share the intuition by
> the reviewer that this may cause important differences. In fact,
> in the literature on stochastic thermodynamics, similar speed limits
> also exist for discrete (Markov) systems; we are planning to address
> this in future work. We will discuss this outlook in the revised manuscript.
>
> Yet, we found that certain features seem to be more universal
> and to carry over between discrete and continuous dynamics; also they
> seem to generalize from the NTK regime (of infinite-sized networks)
> to fully trained finite-sized networks:
> The presented numerical experiments on Myrtle-5 networks in fact employ
> a usual discrete training scheme. Also, we agree that training here is
> not in the NTK regime proper (See also App. Sec. D.). Still, the initial transient of entropy
> production, as explained by the NTK analysis, exists in a qualitatively
> similar manner in the fully trained network. Moreover, the influence of
> the spectrum of the residuals, as predicted by the NTK analysis, correctly
> predicts the transition from the inefficient to the efficient training
> regime.

---

> > ### Author Response · Authors · 2023-11-16
> > **Reply to Weekness: significance point 2:**
> >
> > Significance Point 2:
> >
> > We are thankful for this intriguing question. In fact, given the presented
> > framework in the manuscript, in particular our Eq. (4), and the various
> > existing finite-size theories for deep networks (e.g.,  Li \& Sompolinsky 2020, Zavatone-Veth \& Pehlevan 2021,
> >  Hanin \& Zlokapa 2022, Seroussi \& Ringel 2023), this question can be
> > addressed in a quantitative manner and be studied in relation to different choices of parameters, network architectures etc. Practically, this requires the computation of the irreversibility $R$, given by our Eq. (4) as well as
> > the Wasserstein distance.
> >
> > Both of these quantities can of course readily be measured in numerical experiments;
> > For weak noise, $R$ as given by Eq. (24), is dominated by the gradient and
> > the Hessian of the loss. Likewise, the Wasserstein distance reduces to the
> > Euclidean length in weight space in the noiseless limit.
> >
> > Also from the theoretical side, the reviewer's question is very interesting. We would like to share our preliminary thoughts on this point with the reviewer and are very much interested in
> > their thoughts on this. We feel, however, that a rigorous
> > answer deserves more attention, so we would like to defer to future work.
> >
> > The finiteness of the network and hence the number of
> > parameters $N$ enters the difference of the free energies $F_0 = -\beta^{-1} \ln Z_0$
> > at initialization and $F_T = -\beta^{-1} \ln Z_T$ of the final posterior. These free energies (or partition functions $Z$) are central quantities of many contemporary finite-size network theories (see, e.g., Li \& Sompolinsky 2020, their Fig. (1)).
> > Such frameworks could yield quantitative answers.
> >
> > We here would like to share a preliminary estimate with the reviewer.
> >
> > To compare the quantities for different network sizes, we need to know their scaling with the number of parameters $N$. First note that $W_2$ is extensive in $N$; this is most obvious from our Eq. (25), because the Euclidean squared length of of the thermodynamic velocity $\| v \|^2 = \mathcal{O}(N)$ is proportional to the number of parameters. By the same token, the irreversibility $R$ is extensive in the number of weights, as seen from our Eq. (23).
> > This overall scaling $\propto N$ of the two quantities thus drops out in ratio $W_2/R$ that determines the speed limit. What matters is the irreversibility per degree of freedom
> > and the Wasserstein distance per degree of freedom (per weight).
> >
> > The irreversibility $R$, by our Eq. (4), can likewise be determined from the loss at
> > initialization, and the free energies at initialization and in the final state.
> > We, as well as the reviewer, would like to know if our estimates from kernel methods (such as NTK) provide a lower or an upper bound.
> >
> > First, note that the free energy $F_0/N= -\beta^{-1} \ln Z_0 / N$ per degree of freedom of the random initialization is independent of $N$ and known exactly. The free energy of the posterior $F_T/N = -\beta^{-1} \ln Z_T / N$ depends on $N$ in a complicated manner. The question thus
> > reduces to knowing whether the true $F_T$ is larger or smaller than its mean-field estimate
> > by the NTK. To this end, remember that mean-field approximations can be derived in a
> > variational approach, where one seeks (within a given space of distributions) the distribution
> > that is closest to the true one in terms of the Kullback Leibler (KL) divergence. It is easy to show that from Gibb's inequality (KL divergence being positive semi-definite) it follows that at the minimum one has $0 \le \mathrm{KL} = F_{T, \mathrm{MF}} - F_{T, \mathrm{true}}$. So the mean-field free energy is always larger than or equal to the true one.
> > In terms of the speed limit, this implies that the irreversibility (Eq. (4)) $R = F_0 - F_{T} + \langle \mathcal{L}(0) \rangle$ in the denominator of Eq. (7), estimated by a mean-field method (such as NNGP or NTK), is smaller than the true value. Hence, the speed limit $T_{SL}$ estimated in the mean field is larger than the true value.
> > The previous consideration holds for $W_2/N$, the traveled length in weight space, being the same. So in addition, one would need to analyze how $W_2/N$ changes between the
> > mean-field (NNGP or NTK) approximation and the full weight posterior at finite $N$. We deem
> > an analysis possible by employing contemporary approaches of feature learning (works cited above), but also acknowledge that this highly relevant question deserves more attention to
> > be answered in a satisfactory and rigorous manner. Any further thoughts or advice from the expert reviewer on this issue are of course highly welcome.

---

> > > ### Author Response · Authors · 2023-11-16
> > > **Reply to Weekness-motivation and Questions**
> > >
> > > Motivation:
> > >
> > > The reviewer here precisely asks a highly relevant question.
> > > Yes, indeed, we see one merit of the presented approach in providing guarantees
> > > for the achievable efficiency of the training procedure. But besides these
> > > theoretical guarantees, all involved quantities are readily accessible numerically,
> > > in particular, the irreversibility $R$ given by Eq. (24) being predominantly given
> > > by the local slope and curvature of the loss function, this quantity may be used as
> > > an indicator for periods within the training process that burn most of the "entropy budget";
> > > in the presented numerical experiments and the NTK theory, we identified the initial training
> > > phase as such a critical phase.
> > > Efforts of optimizing the training procedure can hence be targeted to those
> > > phases, reducing the training time most effectively.
> > >
> > > Also, the finding that, depending on the observed power law exponents in the spectrum of
> > > residuals, training is either in the suboptimal or in the optimal regime,
> > > may be used to develop and investigate new forms of initialization that shape
> > > these residuals precisely such that training starts closer to or within the optimal regime.
> > >
> > > Questions:
> > >
> > > Main questions:
> > >
> > > 1. As stated above, we agree that the presented work may be used to evaluate
> > > and assess the closeness to optimality of a given optimizer.
> > > 2. As stated above, some general assertions can be made right away, thanks for the
> > > brilliant question asked by the reviewer, as outlined above. Given the rich
> > > set of finite-size neuronal network theories, and the stepping stone presented here,
> > > namely, exposing a link between the equilibrium free energy and time-bound,
> > > a more fine-grained, network architecture and parameter-dependent analysis is
> > > doable, but requires work beyond the current scope of this manuscript.
> > > 3. We share the assessment by the reviewer that training in the Myrtle-5 network
> > > is outside the NTK regime; also, we did not use Langevin training, but
> > > conventional finite-step gradient descent. This has been done on purpose:
> > > We aimed to see if the theoretical observations in the idealized NTK setting
> > > carry over to the more relevant settings closer to real-world applications.
> > >
> > > Other questions:
> > >
> > > 1. We thank the reviewer for pointing out this indeed unintuitive point,
> > > which we will clarify in the revised version.
> > > We are working with the squared error loss (not the mean squared error loss),
> > > so that gradients are proportional to $n$, the number of training points.
> > > Hence, for constant $\eta$,  the speed of changing weights would
> > > naturally, scale with $n$ in a trivial manner (in practical terms,
> > > training would turn unstable for fixed $\eta$ as $n \to \infty$).
> > > To avoid this trivial "speedup" (and instability) and to instead
> > > focus on the interesting scaling, we need to scale the learning rate
> > > $\eta \propto n^{-1}$. We added an explanation about that in Appendix C.
> > > 2-3. Thanks to the sharp observation by the reviewer, we realize that we failed to communicate this
> > > quite an intricate point, clearly.
> > > The reviewer is of course precisely right that the data distribution is crucial. While for the
> > > analytical results for linear regression data is Gaussian, for the NTK results, which are
> > > required to understand the optimality, the data distribution is more complicated. This distribution
> > > enters only in the scaling form of the NTK kernel's eigenvalues $\lambda_k \propto k^{-\alpha}$ and
> > >  the discrepancies $\Delta^2 \propto k^{-\delta}$.
> > > Such power law distributions are observed throughout many settings (see, e.g., Maloney et al. 2022 https://arxiv.org/abs/2210.16859).
> > > The observation by the reviewer, moreover, is right to the point: While the training time in Fig 1b
> > > is far from optimal (larger than $T_{SL}$ by a factor of about $30$), trajectories are still
> > > somewhat straight (Fig 1d). Frankly, we were quite puzzled at first by this apparent contradiction, too.
> > > The NTK analysis, however, provides an explanation: We find that $T_{SL} \propto T^{1 + \alpha^{-1} (1-\delta)}$. If $\delta>1$, the exponent is negative, so $T_{SL} \ll T$ for large T;
> > > training is hence inefficient. This is what we observe in Fig 1b.
> > > From the geometric point of view, in contrast, the geometric distance $l_{geo}$ and of the distance along the curve $l_\gamma$ both scale identically (Eq. (16)). This implies that the inefficiency we
> > > observe is not due to the motion in a curved space, but rather due to the inhomogeneous speed.
> > > The effect of speed on the entropy production can be appreciated by noting that the entropy
> > > production rate is $ \propto \|v\|^2$, the thermodynamic speed in weight space (Eq. (23)).
> > > We now see that we poorly phrased this point in the initial version, and thank the reviewer for
> > > their helpful remark, which helped us to clarify this point.

---

### Official Review · Reviewer_Lj3X · 2023-12-15

**Soundness:** 2 fair
**Presentation:** 1 poor
**Contribution:** 2 fair
**Rating:** 3
**Confidence:** 5

**Summary:**

The goal of the paper is to explore a notion of training time efficiency in deep learning through the lens of optimal transport and stochastic thermodynamics. The main technical tool is the Benamou-Brenier formula in optimal transport theory, which casts the $L^2$-Wasserstein distance $(W_2$) between initial and final distributions ($\mu_0$ and $\mu_T$) as the minimum of a specific cost functional over smooth paths $(\mu_t)_{0\leq t\leq T}$ induced by time-dependent potentials. The formula provides a lower bound on the transport time (T) in terms of $W_2$ and path-dependent cost.

The authors apply this bound to Langevin dynamics and gradient flow, focusing on linear and kernel regression. They propose to use (a tractable limit of) the bound as a metric for 'training speed inefficiency' in deep learning. Experiments with CNNs trained on small subsets of CIFAR10 with full batch gradient descent on mean squared loss reveal an initial 'inefficient' training phase, corresponding to an alignment of the NTK to the target, followed  by a phase where learning becomes 'efficient,' as evidenced by the training time saturating the proposed bound.

**Strengths:**

The paper introduces an innovative perspective by leveraging the speed limit concept from optimal transport theory to shed light on the training efficiency in deep learning. Given the importance and challenge of characterizing the training speed and efficiency of neural networks,  new theoretical insights in this topic are timely and have the potential for significant impact.

**Weaknesses:**

However, I have significant concerns about the scope and significance of the results, as well as the validity of certain claims. Additionally, I believe there is ample room for substantial improvement in the writing and the clarity of the presentation.

**On Significance**

1. The concept of 'optimal training speed' in the paper is highly specific to the introduced framework and appears to diverge from its conventional interpretation in the optimization literature. Unlike traditional contexts where it refers to achieving optimal convergence rates within a given class of algorithm applied to a specific problem, here, it pertains to conditions on the problem (choice of potential) under which a specific algorithm (Langevin dynamics) aligns with the derived lower bound on transport time (T) from the Benamou-Brenier formula. These two notions of optimality seem orthogonal, explaining potential confusion among reviewers. Reviewer 5tEv rightly points out that the framework doesn't provide suboptimality bounds or convergence rates, hence does not tell us anything about optimality  in the standard sense.  Aligning with Reviewer 8rXj, I believe the relevance to challenges in deep learning optimization remains unclear.


2. Section 2 exposes known results from the literature on the Benamou-Brenier formula (Prop 1 of Benamou-Brenier, 2000; Eq 11 of Vu & Saito, 2022).  The only novel contribution seems to be the simple analytical form (8) of the bound in the noiseless limit (gradient flow), which forms the basis of the NTK analysis in Section 3.2 and the experimental Section 4. However, it appears that the authors may not have explicitly noted that Equation (8) is a direct consequence of the Cauchy-Schwarz inequality:

$$ T \Delta L(T) = - T \int_0^T (\nabla_\theta L)^\top \dot{\theta} dt = T \int_0^T \||\dot{\theta}\||^2 dt \geq \||\theta_T - \theta_0\||^2$$

(where the equality holds for constant $\dot{\theta}$, i.e constant loss gradient along the trajectory).  This raises  questions about the relevance of the speed limit framework invoked by the authors, at least  in the noiseless setting (I will have more comments on the noisy setting of Section 3.1 below).

3. In light of the above, the results in kernel regression (Section 3.2) appear to offer conditions on the target function that minimize the deviation of the flow from a constant gradient loss trajectory (the scale of the actual training time $T$ is fully determined by the kernel spectrum). While these results (along with known results about NTK time evolution and alignment behaviour in the early stage of training, see Fort et al 2020, Baratin et al, 2021,  Atanasov et al 2022) explain empirical observations in Section 4, their broader significance in the context of deep learning optimization remains unclear.

**On Soundness**

4. Some claims and derivations raise validity concerns. For instance, as made explicit above Equ (21) in the appendix, (4) is obtained by choosing $p_T$ to be the stationnary (Boltzman) distribution, reached at infinite time $T \to \infty$. This raises doubts about the validity of the formula at finite times $T$.
2. I have a similar concern for the linear regression analysis of Section 3.1. From Appendix C,  it appears the calculation of the lower bound (7) assumes that $p_T$ is the stationnary distribution (Gaussian centered on the mininum norm solution $\mu_T$)  reached only as $T \to \infty$, limiting the significance of the analysis.  I suspect the same calculation at intermediate finite time $T$ might be more challenging.

5. Section 3.1 statements are puzzling. For example, Eq (12) considers the limit $d\to \infty$ and concludes "interestingly, in this regime, the parameters are not moving a lot and therefore the final distribution is very close to the initial one". But this seems to be merely due to the setting chosen by the authors, where  both initial and target distributions are zero mean Gaussians with O(1/d) variances,  so that $\theta_0 = \theta_\ast = 0$ as $d\to \infty$. Also  the statements following Eq (10), (11), (12) appear misleading, as they refer to a transport saturating the bound,  as opposed to the actual dynamics.

**On Presentation**

The exposition of prior work on speed limits lacks self-containment and clarity.  For example, the formula (3) for entropy production is given without explanation nor intuition. A more effective presentation could start with the variational formulation (27) of the Wasserstein distance from Benamou & Bremier, 2000, build the link with langevin dynamics and gradient flow, and then discuss ways to compute the cost functional. The balance between the main body and the appendix also needs major improvement.  As it stands, the main body compiles (often scattered)  raw results  with (often vague) comments and interpretations, which make it challenging to grasp or appreciate the content without delving into details provided in the supplementary material (or even the prior literature on the topic).

Overall, a (substantially) more integrated presentation might help clarify premises, contributions, and results' implications, making the paper more accessible to the intended audience.

**References**

Fort et al, 2020. Deep learning versus kernel learning: an empirical study of loss landscape geometry and the time evolution of the Neural Tangent Kernel.

Baratin et al, 2021. Implicit Regularization via Neural Feature Alignment

Atanasov et al 2022. Neural Networks as Kernel Learners: The Silent Alignment Effect.

**Questions:**

**Miscalleneous**

* In Eq (7), (27), (28): the Wasserstein distance should be squared.

* Deriving (6) from (5) may not be straightforward, as the term $\nabla_\theta \ln p$ in the integrand of (5) might depend on $\beta$. Additionally, while the derivation of (5) from (41) in Appendix A.6 is acknowledged, the derivation of (41) appears to lack normalization factors (log Z+/Z-) corresponding to the forward and backward distributions. It's not immediately clear to me if these factors cancel each other out.

---

### Meta-Review · Area_Chair_enMz · 2023-12-16

**Metareview:**

This paper uses a statistical thermodynamic perspective to explore lower bounds on the training time of Machine learning models. Considering the entropy generated by the Langevin dynamics and gradient flow through training, the authors developed tools to lower the bounds of the training time of linear regression and neural networks in the NTK regime.


On the one hand, the reviewers received the paper positively. They believe it was a novel perspective to view the optimization trajectory from a thermodynamics standpoint, which will be appreciated by a large enough community of statistical physicists in the ICLR community.

On the other hand, after the discussion, we all agreed that this paper has significant room for improvement. I share the concerns of the reviewers regarding the narrowness of the theoretical setting (all the reviewers), the motivation (Reviewer 8rXj) and the quality of the writing (Reviewer 5tEv). I encourage the authors to:
- Reconsider their title, as their contribution is mainly about the NTK regime. "Speed limits in the NTK regime" would be significantly more accurate.
- add more background on their methods and theory in the appendices,
- be more explicit about what are the contributions and what are already known results along the paper.
- Extend the setting theoretically (going beyond linear models and NTK, studying the discrete update case and the finite horizon case) as well as empirically
- Work on the writing to make it more accessible to an ML audience.



Note: The meta-review above disregards review Lj3X as it was published too late, and the author did not have the chance to rebut that review.
However, I read the paper myself, and I believe that some of the concerns of Reviewer Lj3X are valid. I encourage the author to consider them in their revision. In particular:
- Clarify that Section 2 exposes known results from the literature on the Benamou-Brenier formula (Prop 1 of Benamou-Brenier, 2000; Eq 11 of Vu & Saito, 2022) and that the novelty lies in (8)
- Address the fact that  (8) can be obtained via Cauchy-Swartz inequality without entropy consideration
$$ T \Delta L(T) = - T \int_0^T \langle \nabla_\theta L(\theta(t)), d \theta(t)\rangle =  T \int_0^T \\| \dot \theta \\|^2 dt \geq (\int_0^T \\| \dot \theta \\| dt)^2 \geq \\|\theta(t) - \theta(0)\\|^2\,.$$
- Be clear that the theory only holds for $T \to \infty$

**Justification For Why Not Higher Score:**

The original reviewers appreciated the work but also acknowledged some weaknesses. I still feel very conflicted about this paper as I strongly believe that it is under the ICLR standards for many reasons:
- Rigor of the claims:
  - The theory only holds for $T\to infty$ which is not clear at all in the paper
  - The title is slightly misleading as the authors only consider the linear case and the NTK regime.
- Comparison with related work:
  - Section 2 is listing many previous results and it is not really clear what is known and what is new.
  - No related work section
- Writing:
  - Many results in the main text can only be understood with the appendix
  - notations need to be clearly stated.

**Justification For Why Not Lower Score:**

N/A

---

### Decision · Program_Chairs · 2024-01-16

Reject